EMBO
Molecular Medicine

# Systemic immune challenge exacerbates neurodegeneration in a model of neurological lysosomal disease

Oriana Mandolfo [1], Helen Parker[1,2], Èlia Aguado[1], Yuko Ishikawa Learmonth [1], Ai Yin Liao[1], Claire O'Leary[1], Stuart Ellison[1], Gabriella Forte [1], Jessica Taylor[1], Shaun Wood[1,3], Rachel Searle[1,3], Rebecca J Holley [1], Hervé Boutin[4,5,6] & Brian W Bigger [1,3]✉

## Abstract

**Mucopolysaccharidosis type IIIA (MPS IIIA) is a rare paediatric lysosomal storage disorder, caused by the progressive accumulation of heparan sulphate, resulting in neurocognitive decline and behavioural abnormalities. Anecdotal reports from paediatricians indicate a more severe neurodegeneration in MPS IIIA patients, following infection, suggesting inflammation as a potential driver of neuropathology. To test this hypothesis, we performed acute studies in which WT and MPS IIIA mice were challenged with the TLR3-dependent viral mimetic poly(I:C). The challenge with an acute high poly(I:C) dose exacerbated systemic and brain cytokine expression, especially IL-1β in the hippocampus. This was accompanied by an increase in caspase-1 activity within the brain of MPS IIIA mice with concomitant loss of hippocampal GFAP and NeuN expression. Similar levels of cell damage, together with exacerbation of gliosis, were also observed in MPS IIIA mice following low chronic poly(I:C) dosing. While further investigation is warranted to fully understand the extent of IL-1β involvement in MPS IIIA exacerbated neurodegeneration, our data robustly reinforces our previous findings, indicating IL-1β as a pivotal catalyst for neuropathological processes in MPS IIIA.**

**Keywords** Neurodegeneration; Neuroinflammation; Mucopolysaccharidosis; Sanfilippo; Inflammasome
**Subject Categories** Genetics, Gene Therapy & Genetic Disease; Immunology; Neuroscience

## Introduction

Recurrent viral infections within the respiratory and urinary tract are common in many neurodegenerative diseases (Mitchell et al, 2009; Su et al, 2018), including paediatric neuronopathic lysosomal storage diseases (LSD) (Muhlebach et al, 2011). Among these, it has been estimated that about 88% of Mucopolysaccharidosis (MPS) type IIIA patients suffer from recurrent infections at very early stages in life (Valstar et al, 2010). MPS IIIA is a rare inherited LSD caused by a deficiency of the N-Sulphoglucosamine sulphohydrolase lysosomal enzyme, which is involved in the degradation of heparan sulphate (HS). Chronic accumulation of partially degraded, highly sulphated HS leads to early and progressive neurodegeneration, resulting in rapid cognitive decline, impaired motor skills and severe hyperactivity, with relatively mild somatic symptoms. Currently, no effective disease-modifying therapy is available for MPS IIIA. While ongoing clinical trials are underway, numerous patients find themselves unable to participate in them due to the rapid degeneration associated with the disease (Wijburg et al, 2022).

A more severe cognitive decline in MPS IIIA patients following infection has been anecdotally reported by several paediatricians. In this regard, it is believed that chronic bacterial and viral infections contribute to the progression of several other brain diseases (Perry et al, 2007). Notably, there is evidence linking herpes simplex type 1 (HSV-1) infections with increased risk of Alzheimer's disease (AD)-like pathology development (Eimer et al, 2018), as well as chronic Parkinson-like symptoms (Caggiu et al, 2016, 2017; Agostini et al, 2021), Influenza-A virus (IAV), which also appears to be linked to Parkinson disease onset, was proved to cause cognitive decline and hippocampal morphological changes, when injected into the olfactory bulb of BALB/c mice (Furlan Damiano et al, 2022). In addition, changes in brain structure have also recently been reported in SARS-CoV-2 cases, where reduction in grey matter thickness and changes in markers of tissue damage

[1]Division of Cell Matrix Biology and Regenerative Medicine, Faculty of Biology, Medicine and Health, University of Manchester, 3.721 Stopford Building, Manchester, UK. [2]Lydia Becker Institute of Immunology and Inflammation, School of Biological Sciences, Faculty of Biology, Medicine and Health, University of Manchester, Manchester Academic Health Science Centre, Manchester, UK. [3]Centre for Regenerative Medicine, Institute for Regeneration and Repair, The University of Edinburgh, Edinburgh, UK. [4]Division of Neuroscience & Experimental Psychology, Faculty of Biology, Medicine and Health, University of Manchester, Manchester, UK. [5]Geoffrey Jefferson Brain Research Centre, Manchester Academic Health Science Centre, Northern Care Alliance & University of Manchester, Manchester, UK. [6]INSERM, UMR 1253, iBrain, Université de Tours, Tours, France. ✉E-mail: brian.bigger@ed.ac.uk

were observed in regions that are functionally connected to the primary olfactory cortex (Douaud et al, 2022).

Systemic infections have also been associated with accelerated cognitive decline in AD patients, which is correlated with increased levels of IL-1β in serum (Holmes et al, 2003). IL-1β is an inflammatory cytokine secreted via caspase-1 cleavage, as a result of the NLRP3 inflammasome activation (He et al, 2016).

Increased expression of subunits, activators and downstream mediators of the NLRP3 inflammasome have been detected in MPS IIIA mouse brains (Parker et al, 2020). Furthermore, we have reported that both IL-1β and its receptor antagonist IL-1Ra are elevated in MPS IIIA patients and mouse models, as well as several other MPS diseases (Mandolfo et al, 2022). We have previously demonstrated a two-step model for the activation of the innate immune system in MPS IIIA disease. Notably, MPS IIIA glycosaminoglycans (GAGs) prime pro-IL-1β and NLRP3 transcription and the combination of autophagy dysfunction, lysosomal membrane destabilisation and secondary storage materials mediate NLRP3 activation and IL-1β secretion, eventually leading to exacerbation of the existing inflammatory response (Mandolfo et al, 2022; Parker et al, 2020). IL-1 receptor antagonist (IL-1Ra) competes with IL-1 for binding to IL-1 receptors (IL-1R1 and IL-1R3), and propagates a negative feedback loop. Overexpression of IL-1Ra via haematopoietic stem cell gene therapy resulted in the attenuation of an IL-1 immune response (Parker et al, 2020), which in turn led to reduced microgliosis and astrogliosis in the central nervous system (CNS), as well as complete behavioural correction (Parker et al, 2020). This was corroborated in MPS IIIA x IL-1R1−/− mice, lacking the IL-1R1 receptor, which also resulted in reduced brain glial activation, reversal of working memory deficits and normalisation of hyperactivity, ultimately indicating IL-1 as a key effector of neuroinflammation in MPS IIIA disease (Parker et al, 2020).

Overall, there seems to be a generic mechanism linking systemic infection and cognitive function, whereby infection can exacerbate cognitive impairment in individuals with prior CNS pathology, potentially via IL-1β signalling (Chen et al, 2008; Murray et al, 2012; Lopez-Rodriguez et al, 2021). However, the potential short-term and long-term consequences that viral infections pose on neurological function in children suffering a neurological disease need further elucidation, alongside the mechanisms underlying CNS decline. Here we aimed to investigate this question through the use of the double-stranded RNA analogue and viral mimetic, polyinosinic:polycytidylic acid (poly(I:C)) in a mouse model of MPS IIIA. A number of publications have previously demonstrated that poly(I:C) mimics the acute phase of a viral infection (Fortier et al, 2004; Ratnayake et al, 2014; Murray et al, 2015; Gibney et al, 2013; Weintraub et al, 2014; Traynor et al, 2004).

Poly(I:C) activates toll-like receptor 3 (TLR3) signalling via TRIF-dependent pathways and culminating in the activation of the interferon (IFN) response transcription factor IFN regulatory factor-3 and a number of IFN-β-dependent genes, as well as other pro-inflammatory cytokines in the periphery and CNS (Fernandez-Lizarbe et al, 2009), which are normally involved in the acute symptoms of viral infections.

Here, we have tested both acute and chronic administration of low and high doses of poly(I:C) to mimic, respectively, mild and moderate viral infections (Cunningham et al, 2007; Song et al, 2015) in the mouse model of MPS IIIA. Our investigation has focused on assessing the effects of these treatments on CNS degeneration, systemic/neuroinflammation, and behaviour in the context of this chronic neuronopathic disease.

# Results

## Acute administration of poly(I:C) induces a sickness behaviour response

Poly(I:C) is known to induce a dose-response cytokine-dependent sickness behaviour in mice, along with hypoactivity, weight loss and hyperthermia at 6 h (Murray et al, 2015; Fortier et al, 2004; Traynor et al, 2004; Cunningham et al, 2007; Blank et al, 2016). In order to assess that the mice were responding to poly(I:C), 2- to 4-month-old WT and MPS IIIA mice were administered either saline or poly(I:C), and the sickness behaviour response was evaluated. In this respect, two doses were compared: a high dose (12 mg/kg, administered intraperitoneally), which mimics a moderate infection (Fig. 1A) and a low dose (6 mg/kg, administered intravenously) which mimics a mild infection (Fig. 1D).

At 6-h post-administration, both WT and MPS IIIA groups challenged with high dose (12 mg/kg) poly(I:C) displayed a significant weight loss (13.6% and 18.2%, respectively) (Fig. 1B), whilst both groups of mice treated with the low poly(I:C) dose (6 mg/kg), also showed significant 6% body weight reduction (Fig. 1E). Interestingly, at 24-h post-administration, the poly(I:C) treated groups retained a significant reduction in body weight when compared to their saline-treated controls, likely as a result of induced sickness behaviour.

Poly(I:C) also induced significant reductions in burrowing performed 5–7 h post challenge. Burrowing activity was significantly elevated in saline-treated MPS IIIA mice compared to WT mice, which is in line with the hyperactive phenotype observed in MPS IIIA animals (Langford-Smith et al, 2011). High-dose poly(I:C) treatment reduced burrowing activity by 39% in WT animals, and 72% in MPS IIIA animals vs their respective saline group (Fig. 1C). When subjected to low-dose poly(I:C) treatment, WT mice exhibited a tendency towards diminished burrowing activity; however, this trend did not achieve statistical significance. (Fig. 1F). Similarly, MPS IIIA mice treated with low-dose poly(I:C) displayed a significant 85% reduction when compared to their saline control (Fig. 1F). These data demonstrate that poly(I:C) induces a dose-dependent sickness behaviour in WT and MPS IIIA mice, which persists past 24 h following administration.

## An acute high poly(I:C) dose induces mild microgliosis in WT and a significant decrease in GFAP[+] astrocytes and NeuN[+] neurons in MPS IIIA mice

In order to understand if an acute high poly(I:C) dose administration was able to activate glia and cause neuronal loss in the MPS IIIA brain, we stained histological brain sections with Isolectin-B4 (ILB4) and Glial fibrillary acidic protein (GFAP), which are markers of active microglia and astrocytes, respectively. Overall, significantly more ILB4 and GFAP staining was displayed in the hippocampus of saline-treated MPS IIIA mice when compared to saline-treated WT mice, indicative of extensive gliosis.

Challenge with an acute high poly(I:C) dose led to significant microglial activation in the hippocampus of WT animals 3 h post

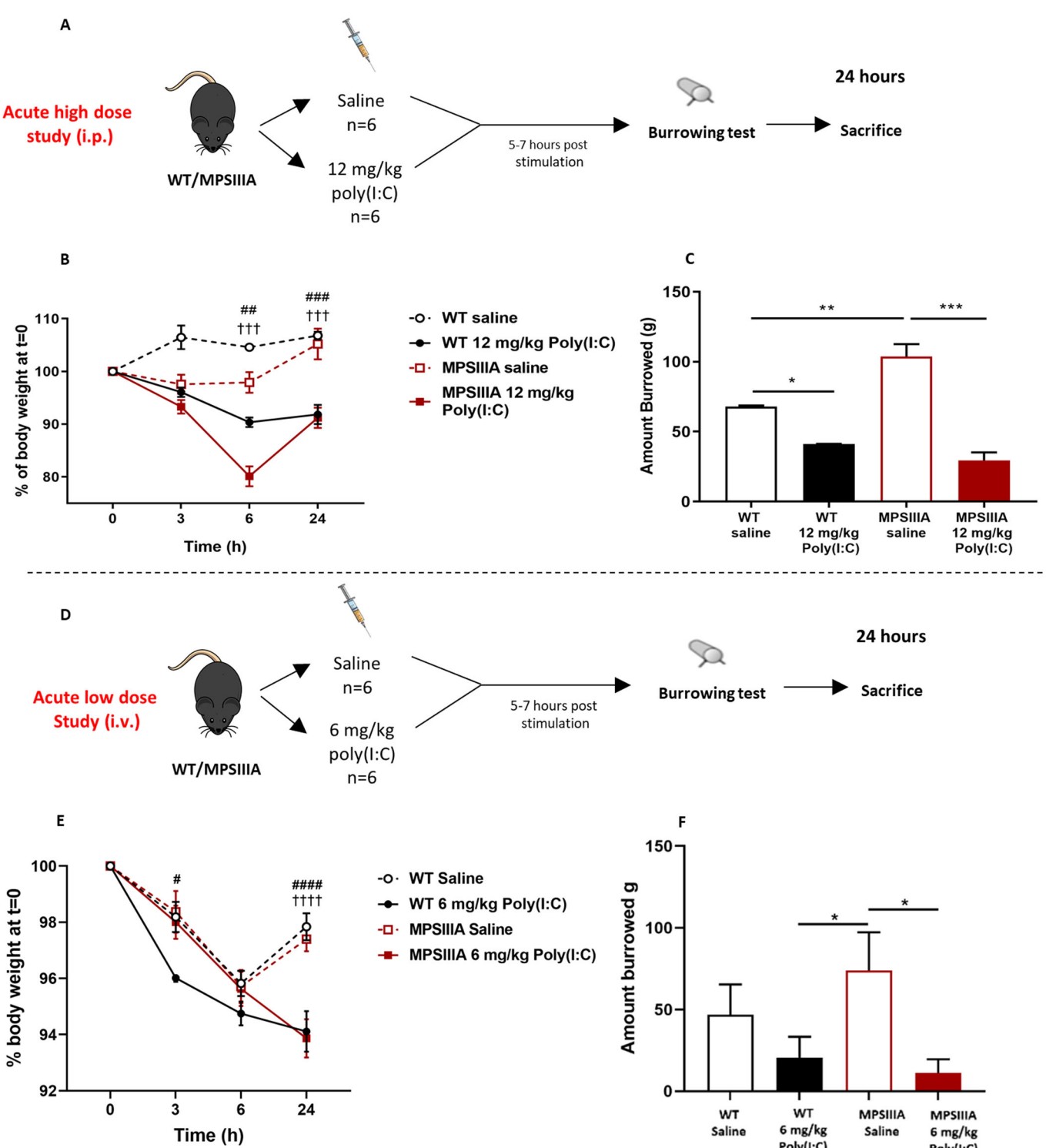

poly(I:C) challenge, but no further exacerbation of the existing microgliosis in MPS IIIA animals (Fig. 2A). With regards to astrogliosis, an acute high poly(I:C) dose induced significant astrocyte activation in the hippocampus of WT mice, 3 h post challenge, with resolution evident by 24 h (Fig. 2B). Strikingly, while poly(I:C) challenge had no additional effect on pre-existing astrogliosis in MPS IIIA mice at 3 and 6 h post challenge, there was

a contrasting sharp reduction in GFAP$^+$ astrocytes in the MPS IIIA hippocampus at 24 h post challenge, aligning with levels observed in WT mice. Notably, changes in apparent astrocyte levels were exclusive to the hippocampus following poly(I:C) treatment.

Since microglial activation and secretion of neurotoxic factors can lead to neuronal cell death (Depino et al, 2005), we then used NeuN staining to examine the effect of an acute high poly(I:C)

**Figure 1. Acute administration of poly(I:C) induces a sickness behaviour response.**

(A) In the acute high-dose study, 6 cohorts of 4 month old female WT or MPS IIIA mice ($n = 6$) were administered a single challenge of saline or 12 mg/kg poly(I:C). (B) The cohorts were measured for body weight and sacrificed at 3, 6 and 24 h post injection by an intra-cardiac perfusion of PBS under terminal anaesthesia ($n = 6$). Body weight is presented as a % of body weight at $t = 0$. The three saline controls were combined ($n = 18$). (C) The burrowing behaviour was assessed between 5 and 7 h post stimulation in the 6 and 24 h cohorts ($n = 6$). (D) In the acute low-dose study, 2 to 4 month old WT and MPS IIIA mice were administered a single challenge of saline or 6 mg/kg poly(I:C). (E) Body weight was measured at the time of poly(I:C) or saline administration ($t = 0$) and again at 3, 6 or 24 h post poly(I:C) or saline ($n = 6$). Body weight is presented as a % of body weight at $t = 0$. (F) Burrowing was assessed between 5 and 7 h post poly(I:C) or saline challenge in WT and MPIIIA animals ($n = 6$). Error bars represent the standard error of the mean (SEM). Significant differences are determined by two-way ANOVA with Tukey's post hoc analysis. Comparisons showing significant differences are denoted by *$P < 0.05$, **$P < 0.01$ and ***$P < 0.001$ above comparison lines. # indicates poly(I:C) being significantly different from saline for WT animals; † indicates poly(I:C) being significantly different from saline for MPS IIIA animals, whereby # or †$<0.05$, ## or ††$P < 0.01$, ### or †††$P < 0.001$ and #### or ††††$P < 0.0001$ above comparison lines. Source data are available online for this figure.

dosing on neuronal loss 24 h after administration. Saline-treated MPS IIIA mice already display a significant decrease in NeuN+ neurons over WT saline-treated controls; however, while poly(I:C) had no effect on neuronal viability in the hippocampus of WT animals, it significantly exacerbated the existing decrease in hippocampal NeuN+ neurons in MPS IIIA mice (Fig. 2C).

## An acute high poly(I:C) dose exacerbates Il-1β levels and caspase-1 activity in the serum and brain of MPS IIIA mice

Inflammasome activation plays a role in the host defence against viral infection, whereby it activates caspase-1 and initiates the release of IL-1β (Rajan et al, 2010; Poeck et al, 2010). Caspase-1 activation can further activate the inflammasome and cause pyroptosis-mediated cell death. We have previously shown that IL-1β and the NLRP3 inflammasome are upregulated in MPS IIIA mice (Parker et al, 2020). In order to determine if poly(I:C) could further exacerbate this phenotype, we examined peripheral and brain IL-1β and other inflammatory markers, following challenge with either a single high dose or single low-dose poly(I:C).

Challenge with an acute high poly(I:C) dose induced a significant 3.6 fold and 8.2 fold increase in IL-1β peripheral levels in MPS IIIA and WT animals respectively, at 6 h post challenge, when compared to their respective saline control groups. Interestingly, IL-1β remained significantly upregulated by 2.4 fold in MPS IIIA mice when compared to WT animals 24 h post poly(I:C) (Fig. 3A). An acute high poly(I:C) dose also induced a significant increase in TNF-α peripheral levels in both WT (2.4 fold) and MPS IIIA (2.3 fold) animals 3 h post challenge, when compared to their respective saline control groups, with normalisation at 24 h (Fig. 3B). The direct impact of peripheral cytokines and poly(I:C) on cytokine expression in the brain remains elusive. To investigate the inflammatory response triggered by poly(I:C) within the brain, we evaluated the mRNA expression of pro-inflammatory cytokines—IL-1β, TNF-α and MCP-1—in whole brains of WT and MPS IIIA animals at 3, 6 and 24 h post challenge with 12 mg/kg poly(I:C). Poly(I:C) induced a pronounced 5.8 fold increase in *Il1b* gene expression in MPS IIIA animals 3 h post challenge, compared to WT animals in the same treatment group (Fig. 3C). Furthermore, *Il1b* expression remained significantly upregulated at 6 and 24 h post poly(I:C) in comparison to both WT animals at the respective timepoints and MPS IIIA saline controls. *Tnfa* exhibited a significant upregulation 3 h after poly(I:C) treatment in both WT and MPS IIIA animals, returning to baseline levels for both groups

at 24 h (Fig. 3D). In contrast, no upregulation in *Ccl2* gene expression was observed in either poly(I:C)-treated group (Fig. 3E).

Motivated by these findings, we explored the potential upregulation of IL-1β protein expression in the CNS of MPS IIIA mice. No histological evidence of IL-1β positive cells was observed in the brain of saline- or poly(I:C)-challenged WT mice (Fig. 3F). Conversely, while IL-1β cells were sporadically seen throughout the hippocampus of saline-treated MPS IIIA animals, a further increase in IL-1β staining was detected following challenge with poly(I:C). In agreement with this observation, the levels of caspase-1 activity, which is involved in processing pro-IL-1β into mature IL-1β, were also significantly increased in whole brain homogenates of MPS IIIA mice, when compared to saline-treated WT animals (Fig. 3G), suggesting increased inflammasome activation. Furthermore, poly(I:C) exacerbated caspase-1 activity 6 h post challenge in both WT and MPS IIIA animals when compared to their respective saline-treated groups.

Following acute challenge with a low dose of poly(I:C), no significant exacerbation in IL-1β expression was observed in the periphery at 24 h post-administration (Fig. 3H), although the assay was close to the limit of detection. Conversely, a significant 2.1 fold increase in peripheral TNF-α was detected in the treated MPS IIIA animals (Fig. 3I). This trend extended to the brain for both cytokines; specifically, while the low dose of poly(I:C) had no impact on *Il1b* gene expression (Fig. 3J), it did result in a significant 3.6 fold increase in *Tnfa* gene expression in WT mice (Fig. 3K). Furthermore, a 4- and a 2.2-fold increase in brain *Ccl2* gene expression was also observed in the poly(I:C)-treated WT and MPS IIIA, respectively, when compared to their corresponding saline controls (Fig. 3L). When we explored the potential upregulation of IL-1β protein expression in the CNS, we detected basal levels of microglial IL-1β expression in the hippocampus of WT mice, showing no significant exacerbation after low acute poly(I:C) treatment (Fig. 3M). Conversely, MPS IIIA animals exhibited a trend towards increased IL-1β+ cells in the hippocampus, further exacerbated in the poly(I:C)-treated group (Fig. 3N).

## An acute low poly(I:C) dose induces an increased monocyte-driven innate immune response in the spleen of MPS IIIA mice

Based on the elevations of MCP-1 identified in plasma and brain after an acute low poly(I:C) dose, we decided to investigate the production and activation of myeloid cells in the spleen, in response to low-dose poly(I:C) delivery and how these change in MPS IIIA using flow cytometry. Overall, both WT and MPS IIIA saline-treated groups displayed similar baseline levels of innate immune cells (Fig. 4). Challenge with poly(I:C) induced an

# Acute high dose study (i.p.)

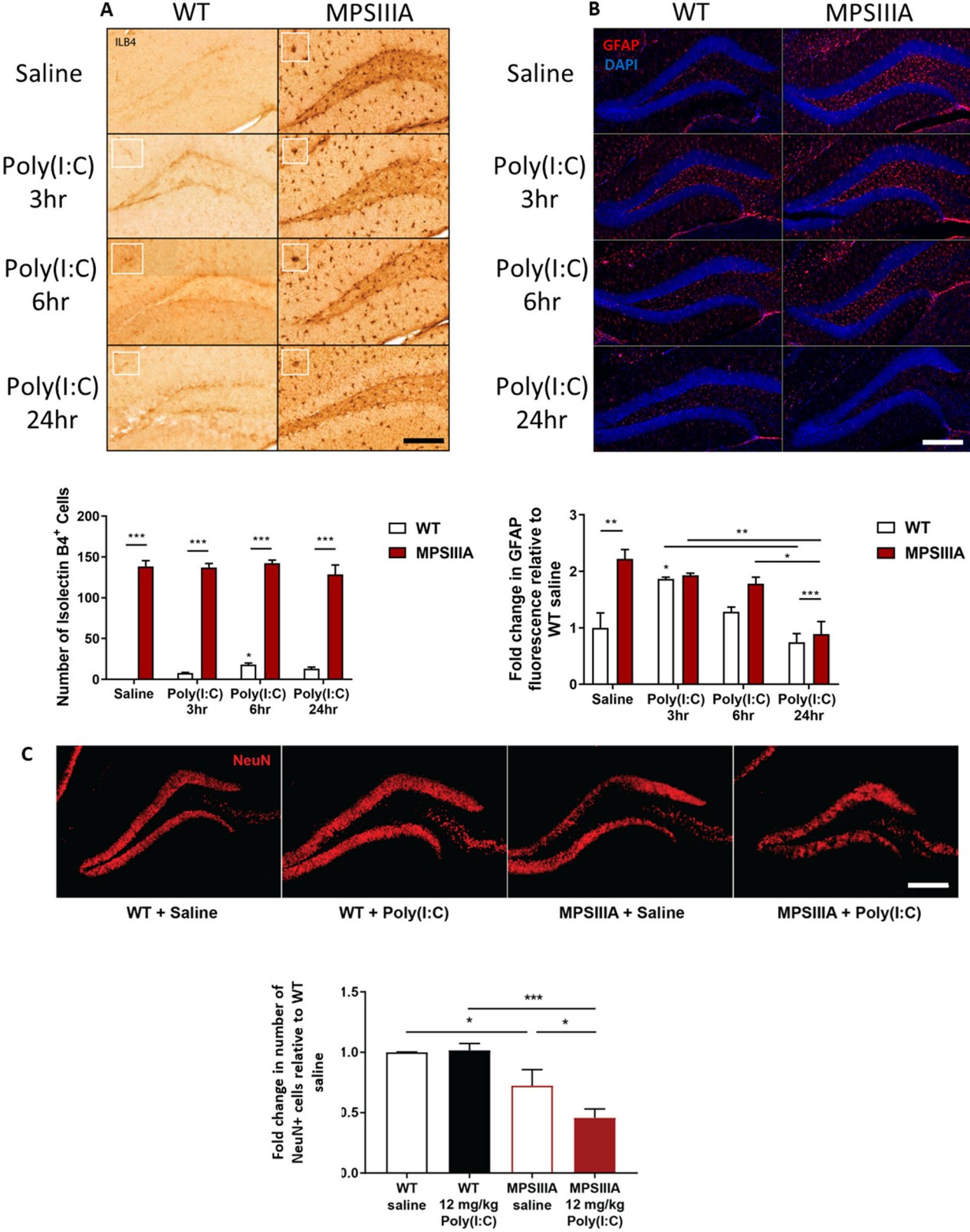

**Figure 2. An acute high poly(I:C) dose induces mild microgliosis in WT and a significant decrease in astrogliosis, and increase in neuronal loss in MPS IIIA mice.**

Microglial and astrocyte activation was assessed 3, 6 and 24 h after poly(I:C) or saline in WT and MPS IIIA hippocampi via isolectin-B4 labelling and GFAP immunoreactivity, respectively (**A, B**). Scale bar = 200 μm. Insets demonstrate representative microglia. The number of ILB4+ cells was manually quantified in ImageJ. The level of GFAP fluorescence was quantified in ImageJ, and normalised against background measurements and GFAP fluorescence detected in saline-treated WT animals. (**C**) Neuronal loss was assessed 24 h after poly(I:C) or saline in WT and MPS IIIA hippocampi via NeuN labelling. The number of NeuN+ cells was quantified in ImageJ. Scale bar = 100 μm. Error bars represent standard error of the mean (SEM), $n = 3$ in all groups. Significant differences were determined by two-way ANOVA with Tukey's post hoc analysis, *$P < 0.05$, **$P < 0.01$ and ***$P < 0.001$. Significance against WT saline is shown above recorded data, all other comparisons are shown with a line. Source data are available online for this figure.

expansion of all cell types in all treatment groups, with similar levels of neutrophils and dendritic cells (DCs) detected in poly(I:C)-treated WT and MPS IIIA mice (Fig. 4A–C). Strikingly, a more exacerbated trend was observed with regard to monocytes and macrophages in the poly(I:C)-treated MPS IIIA mice compared to poly(I:C)-treated WT mice. In this respect, a 17% and a significant 78% increase in CD86+ monocytes were observed in WT and MPS IIIA mice, respectively, following poly(I:C) challenge (Fig. 4D), when compared to their saline control groups. Furthermore, monocytes not only became more active following poly(I:C) administration, but also showed to proliferate more, as a significant increment of about 2% was observed in total monocytes (% Ly6C+ monocytes of total CD45+ cells) in both treatment groups (Fig. 4D). A similar trend was also observed in macrophages, with an increment in the activation rate of 24.6% in the poly(I:C)-treated WT group, and a significant increment of 53.1% in the poly(I:C)-treated MPS IIIA group, when compared to their saline controls (Fig. 4E).

## Chronic administration of poly(I:C) adversely affects MPS IIIA behaviour in a dose-dependent manner

Chronic poly(I:C) dosing has also been shown to impair synaptic plasticity and long-term potentiation, to affect memory consolidation by disrupting hippocampal-dependent learning and memory and to confer anxiety-like behaviours in neonatal animals (Perry et al, 2007; Palin et al, 2008; Barrientos et al, 2006). In order to test the hypothesis that recurrent viral infections can accelerate the existing neurodegeneration in MPS IIIA patients, 2-month-old WT and MPS IIIA mice were treated with either saline or increasing doses of poly(I:C) and the impact on cognition was evaluated through y-maze spontaneous alternation test (Fig. 5). In this respect, the impact of repeated moderate chronic challenges from high-dose poly(I:C) (6, 9, 12 mg/kg; administered intraperitoneally) (Fig. 5A) was compared against the effects caused by repeated mild chronic challenges from low-dose poly(I:C) (2.5, 4, 6 mg/kg; administered intravenously) (Fig. 5C).

Following each poly(I:C) challenge, sustained reduced body weight, decreased burrowing activity and increased blood MCP-1 chemokine were detected in both treatment groups, suggesting successful poly(I:C) injection (Fig. EV2).

In the chronic high-dose study, a significant 23% deficit in working memory was observed in MPS IIIA saline control mice when compared to WT animals. Repeated administration of poly(I:C) further impaired this deficit in working memory of MPS IIIA animals by 29%, when compared to saline-treated MPS IIIA animals. Conversely, repeated administration of high chronic poly(I:C) doses had no impact on the spatial working memory of WT animals (Fig. 5B), when assessed at 12 weeks after the first injection.

In the chronic low-dose poly(I:C) study, spatial working memory was assessed at 8, 12 and 16 weeks after the first injection (Fig. 5D–F). Overall, a similar trend characterised by a reduction in spatial working memory functioning in both MPS IIIA groups and poly(I:C)-treated WTs was observed across all timepoints, but only 16 weeks after the first injection, a significant 26% deficit in working memory was observed in MPS IIIA saline control mice when compared to WT animals. However, repeated chronic low-dose poly(I:C) treatment did not exacerbate this further.

## Chronic poly(I:C) leads to increased significance of expression of several key pro-inflammatory genes in the CNS of MPS IIIA mice

Overall, our data suggest extensive brain damage induced by high chronic poly(I:C) dosing similar to a moderate infection in MPS IIIA mice. However, because we had already seen significant neuronal dropout from high-dose acute challenges, as well as worsening behaviour from chronic high-dose challenges, we sought instead to understand if neuropathological changes were induced by low chronic poly(I:C) dosing, even though this treatment did not have an impact on the MPS IIIA behavioural phenotype.

The direct impact of peripheral cytokines and poly(I:C) on the expression of cytokines in the brain remains unclear. To investigate the inflammatory response within the brain to repeated low poly(I:C) doses, mRNA expression of pro-inflammatory and anti-inflammatory cytokines, as well as inflammasome components were assessed in the brains of WT and MPS IIIA animals following poly(I:C) challenges. Overall, all inflammatory markers, including components of the NLRP3 inflammasome (*Tnfa, Il1b, Ccl3, Tlr3, Il1ra, Ccl2, Nlrp3* and *Pycard*) were significantly upregulated in saline-treated MPS IIIA mice, when compared to saline-treated WT animals (Fig. EV3A–H). No upregulation was detected in WT mice about 2 months after the last poly(I:C) injection, in contrast with what observed at 24-h post injection, suggesting that these cytokines are only transiently upregulated immediately after poly(I:C) dosing. Treatment with poly(I:C) did not significantly increase any of these inflammatory factors in MPS IIIA mice, but it did increase the significance of the response in the case of *Nlrp3* and *Tlr3*, with trends to an increase in *Tnfa* and *Il1b*.

## Chronic low-dose poly(I:C) administration leads to increased microgliosis in the cortex and amygdala of MPS IIIA mice

Maintenance of the neuron-microglia-astrocyte 'triad' is critical for normal brain function, yet extensive gliosis is a marker of MPS IIIA disease (Wilkinson et al, 2012). Following the observance of increased MCP-1, in

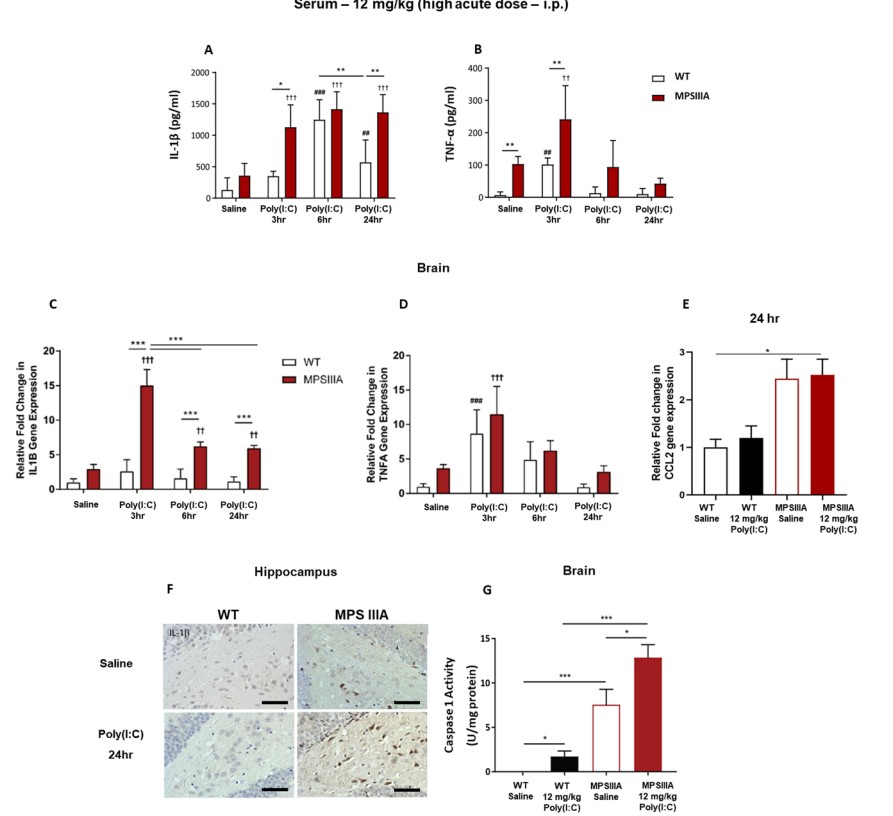

**Serum – 12 mg/kg (high acute dose – i.p.)**

**Brain**

**Hippocampus**

**Brain**

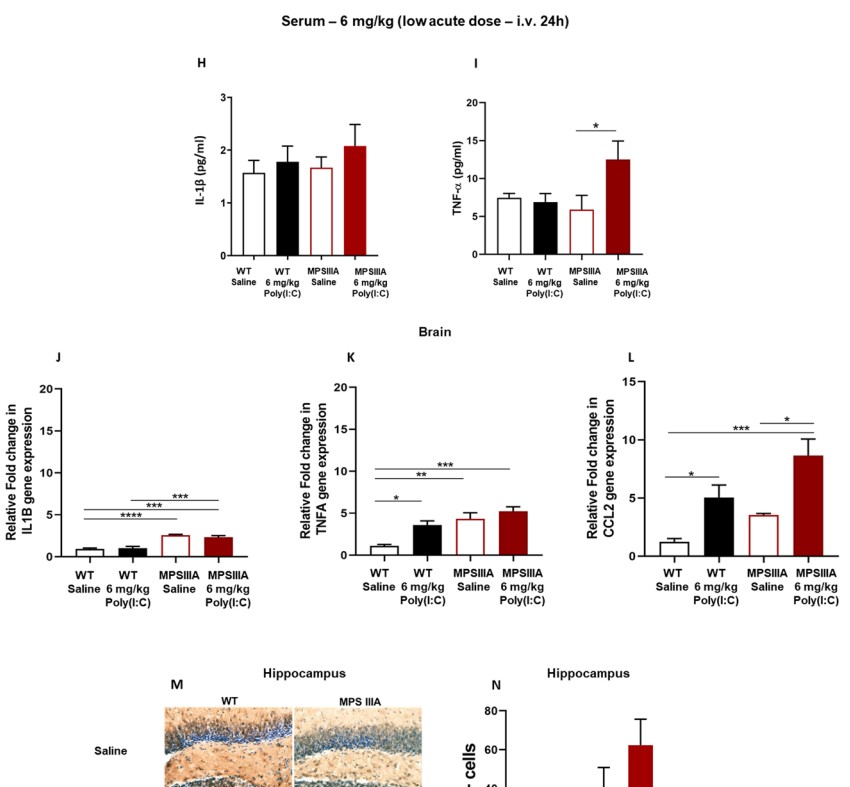

**Serum – 6 mg/kg (low acute dose – i.v. 24h)**

**Brain**

**Hippocampus**

**Hippocampus**

**Figure 3.   An acute high poly(I:C) dose exacerbates Il-1β levels and caspase-1 activity in the serum and brain of MPS IIIA mice.**

Levels of IL-1β (**A**) and TNF-α (**B**) were analysed in serum, collected at various times after challenge with saline or 12 mg/kg poly(I:C) (3-, 6- and 24-h post challenge) and evaluated via ELISA ($n = 6–18$). mRNA levels of *Il1b* (**C**), *TNFa* (**D**) and *Ccl2* (**E**) were analysed in the brain collected at 3, 6 and 24 h post challenge with saline or 12 mg/kg poly(I:C) ($n = 6$) and evaluated via qPCR. (**F**) Intracellular IL-1β expression was assessed 24 h after 12 mg/kg poly(I:C) or saline in WT and MPS IIIA hippocampi. Scale bar = 50 μm ($n = 6$). (**G**) Caspase-1 activity within whole brain homogenates was determined via a fluorescent enzyme assay and normalised against total amounts of protein ($n = 6$). Levels of IL-1β (**H**) and TNF-α (**I**) were analysed in serum collected at 24 h after challenge with saline or 6 mg/kg poly(I:C) ($n = 3$) and evaluated via ELISA. mRNA levels of *Il1b* (**J**), *TNFa* (**K**) and *Ccl2* (**L**) were analysed in the brain collected at 24 h after challenge with saline or 6 mg/kg poly(I:C) ($n = 6$) and evaluated via qPCR. (**M**) Intracellular IL-1β expression was assessed 24 h after 6 mg/kg poly(I:C) or saline in WT and MPS IIIA hippocampi. Scale bar = 50 μm ($n = 4$). (**N**) The number of IL-1β+ cells in the hippocampus was manually quantified using ImageJ ($n = 3–4$). Significant differences are determined by two-way ANOVA with Tukey's post hoc analysis. Error bars represent standard error of the mean (SEM). Comparisons showing significant differences are denoted by $*P < 0.05$, $**P < 0.01$, $***P < 0.001$ and $****P < 0.0001$ above comparison lines. # indicates poly(I:C) being significantly different from saline for WT animals; † indicates poly(I:C) being significantly different from saline for MPS IIIA animals (## or ††$P < 0.01$ and ### or †††$P < 0.001$). Source data are available online for this figure.

both acute and chronic low poly(I:C) dosing, as well as the increased production of splenic monocytes and macrophages in the chronic disease setting of MPS IIIA, we sought to determine if this translated into long-term elevation of microglial cell engraftment and activation in the brain. Significantly more ILB4 staining was displayed in untreated MPS IIIA than in WT mice in the cortex, hippocampus and amygdala, indicative of extensive microgliosis (Fig. 6A). Overall, long-term exposure to poly(I:C) did not induce any microglia activation in WT mice. Conversely, poly(I:C) treatment induced a significant increase in the total number of ILB4+ cells in both cortex (13.5%) and amygdala (12.4%) (Fig. 6B,D) of the MPS IIIA mice, but not in the hippocampus (Fig. 6C).

## Chronic low-dose poly(I:C) administration exacerbates astrogliosis in the cortex and amygdala but leads to GFAP+ astrocyte reduction in the hippocampus of MPS IIIA mice

Elevated astrogliosis is also a pathological feature of several MPS diseases, including MPS IIIA (Wilkinson et al, 2012), hence, we also examined the impact of long-term exposure to low doses of poly(I:C) on astrocyte activation. Brain coronal sections of control and treated MPS IIIA mice were stained with the astrocytic marker GFAP (green). Overall, significantly more GFAP staining was observed in untreated MPS IIIA than in WT mice in the cortex, and amygdala, indicative of extensive astrogliosis (Fig. 7A). Conversely, only a minor increase was observed in the hippocampus, as nearly all astrocytes in the healthy hippocampus are known to express detectable GFAP (Khakh and Sofroniew, 2015). Following repeated poly(I:C) challenges, no difference was observed in astrocytic activation in the cortex of WT mice (Fig. 7B), yet an increasing trend was detected in the amygdala (Fig. 7D). With regards to the MPS IIIA group, poly(I:C) induced a significant exacerbation in the total number of GFAP+ cells in both cortex (21.3%) (Fig. 7B) and amygdala (23.5%) (Fig. 7D), when compared to the saline control. Strikingly, when analysing the hippocampus, poly(I:C) induced downregulation of GFAP+ astrocytes in both WT and MPS IIIA mice (Fig. 7C); notably, a 26.5% and 17.3% decrease in GFAP+ cells was observed in poly(I:C)-treated WT and MPS IIIA mice respectively, when compared to their saline controls.

## Chronic low-dose poly(I:C) administration leads to a reduction in NeuN+ neurons in the hippocampus of MPS IIIA mice

Since exposure to a high single poly(I:C) dose induced a significant reduction in the number of NeuN-expressing neurons in MPS IIIA mice when compared to the saline control, we investigated whether

a similar effect is achieved following repeated challenges with low poly(I:C) doses. Brain coronal sections of control and treated MPS IIIA mice were stained with the neuronal marker NeuN (green) (Fig. 8A). Overall, although a decreasing trend was observed in the total number of NeuN+ cells in the cortex of poly(I:C) treated WT and both MPS IIIA groups, when compared to saline-treated WT mice, no significant difference was detected across groups (Fig. 8B). With regards to the hippocampus, no difference was observed in the percentage of NeuN positive area in the WT mice challenged with poly(I:C), compared to the saline control (Fig. 8C). Strikingly, a 17.1% decrease in percentage of NeuN+ area ($P = 0.04$) was detected in the MPS IIIA mice challenged with poly(I:C), when compared to mice treated with saline.

## Discussion

We hypothesised that in a chronic neurodegenerative disease such as MPS IIIA, systemic infections could exacerbate existing symptoms and drive the progression of neurodegeneration. To test this hypothesis, we utilised poly(I:C), a double-stranded RNA (dsRNA) product serving as a viral mimetic that induces a type I interferon (IFN-I) response. In vertebrates, the initiation of the type I interferon (IFN) response is triggered by dsRNA, a by-product of viral replication. It engages receptors such as RIG-I, RNA helicase A/DHX9, MDA5, TLR3, and SR-As, initiating a signalling cascade involving adaptor proteins like IRF3/7, leading to IFN production (Poynter and DeWitte-Orr, 2018). While a real viral infection triggers both innate and adaptive immune responses, involving B cells, T cells, antibody production, and immunological memory, poly(I:C) solely induces an innate immune response. Recognised by endosomal TLR3 and RIG-I-like receptors (RLR) like RIG-I and MDA5, poly(I:C) activates NFκB and IRF3, resulting in the upregulation of IFN-I, cytokines, and dendritic cell maturation (Mcgarry et al, 2021). IFN-I responses induced by both viral infections and poly(I:C) contribute to the systemic symptoms observed in acute viral infections, such as fever and reduced locomotor activity (Traynor et al, 2004).

In this study, we compared the effects of poly(I:C) at low (6 mg/kg) and high (12 mg/kg) doses to model mild and moderate infections, respectively, in both WT and MPS IIIA mice. These doses were chosen based on previous dose-response studies (Cunningham et al, 2007), in which 2, 6 and 12 mg/kg poly(I:C) induced a dose–responsive sickness behaviour, characterised by decreasing locomotor activity, burrowing and body weight, and

# Acute low dose study (i.v)

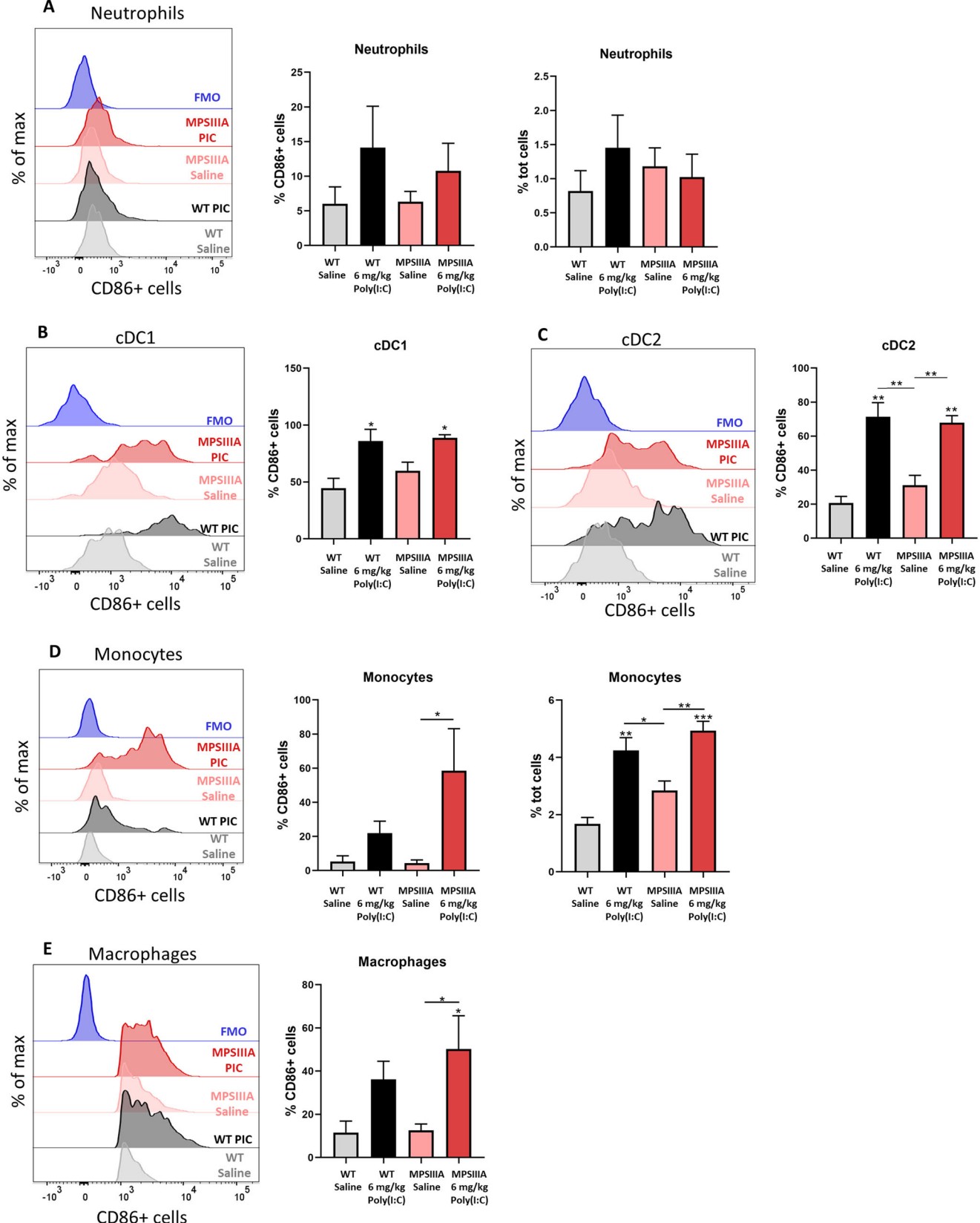

◄  **Figure 4.  An acute low poly(I:C) dose induces an increased monocyte-driven innate immune response in the spleen of MPS IIIA mice.**

% of CD86+ neutrophils (**A**), cDC1 (**B**), cDC2 (**C**), monocytes (**D**) and macrophages (**E**) in whole spleen of 2–4 month old WT and MPS IIIA animals (24 h post challenge) receiving a single treatment of saline or 6 mg/kg poly(I:C). The % of CD86+ cells was measured by flow cytometry and further analysed with FlowJo v10. Fluorescence minus one (FMO) controls are displayed. % of neutrophils (**A**) and monocytes (**D**) compared to % total cells in the whole spleen was also analysed with FlowJo v10. Significant differences are determined by two-way ANOVA with Tukey's post hoc analysis. Comparisons showing significant differences are denoted by *$P < 0.05$, **$P < 0.01$ and ***$P < 0.001$ above comparison lines. Significance against WT is shown above recorded data. Error bars represent standard error of the mean (SEM) ($n = 3$). Source data are available online for this figure.

hyperthermia. Furthermore, 12 mg/kg poly(I:C) proved to induce significant hypothermia and weight loss in normal mice at later times, that is comparable with that caused by influenza virus at 0.1 of its $LD_{50}$ (Song et al, 2015). In addition, we also performed a further dose-response study, in which WT mice were injected with 6, 9 and 12 mg/kg poly(I:C) via three different routes (IP, IV and subcutaneous). Our comparative study indicated a similar sickness behaviour and immunogenic response when administering 12 mg/kg poly(I:C) IV, or IP and lower responses with 6 mg/kg IV. This is provided in (Fig. EV4). From our data, it is clear that all poly(I:C) dosing strategies effectively induced innate immune responses and sickness behaviour in both WT and MPS IIIA animals.

In our study, serum IL-1β remained significantly upregulated in the MPS IIIA mice when compared to WT, after recovery from an acute high poly(I:C) dose. This was correlated with increased IL-1β protein expression in the hippocampus, which may be localised to microglia-like cells, as well as exacerbation of caspase-1 activity and hippocampal cell damage, all indicative of NLRP3 activation. This is in agreement with previous studies that showed that systemic infection models, including the TLR3 viral mimetic poly(I:C), are able to induce NLRP3 inflammasome activation via RIG-1/MDA5/MAV-dependent and -independent pathways, whereby secretion of IL-1β via caspase-1 activation is potentiated (Rajan et al, 2010), eventually leading to pyropoptosis and lysis of the affected cell (Pandey and Zhou, 2022; Sefik et al, 2022; Karaba et al, 2020). Indeed, evidence of astrocytic and neuronal cell damage was present in the hippocampus of the MPS IIIA mice, following an acute high poly(I:C) dose. When administering an acute low poly(I:C) dose, increasing trends were still observed in brain IL-1β, with a significantly exacerbated MCP-1-mediated inflammatory response in MPS IIIA plasma and brain. MCP-1 is reported to be upregulated following both TLR3 and TLR4 activation in macrophages, with the primary function of activating and inducing the migration of leukocytes (Pattison et al, 2013). This is in line with our findings, showing a more exacerbated activation of both monocytes and macrophages in the MPS IIIA spleen, following an acute low systemic challenge. Nevertheless, while macrophage activation was observed systemically, no worsening in the existing microgliosis was observed in the hippocampus of MPS IIIA mice after an acute high poly(I:C) dose.

Importantly, when exposed to chronic low poly(I:C) doses, long-term elevation of reactive microglia was detected in both MPS IIIA cortex and amygdala, as opposed to WT mice in which microgliosis was not induced. In this respect, we hypothesised that recurrent flu-like infections are likely to generate a hyper-activated microglia response in MPS IIIA, characterised by inflammasome activation. This can either lead to proliferation of resident microglia or to increased recruitment of peripheral monocytes to the brain, where they ultimately exacerbate the pre-existing inflammatory response

characterising the disease (Godbout et al, 2005), similarly to what is observed in AD patients (Holmes et al, 2003).

Similarly to our acute high poly(I:C) study, neuronal perturbation was also observed in the hippocampus of MPS IIIA mice treated with chronic low poly(I:C) doses. While reactive microglia seem to be responsible for neuronal degeneration in several neuropathic LSDs (Ko et al, 2005; Elrick et al, 2010; Parviainen et al, 2017), we did not observe exacerbated microgliosis in the hippocampus of MPS IIIA mice following chronic low poly(I:C) doses. However, increased IL-1β activity in MPS IIIA mice (and MPS IIIA mice treated with poly(I:C)) seems to primarily stem from microglia (Fig. 3M,N). This suggests that the IL-1β-driven inflammation and behavioural abnormalities previously observed in MPS IIIA are likely mainly driven by macrophage/microglial IL-1β (Parker et al, 2020). Notably, neuronal degeneration can not only be attributed to increased microgliosis, but also to any potential inflammatory factors released as a result of astrocytic atrophy or weakening of neuronal synapses (Verkhratsky et al, 2016). Interestingly, a reduction in GFAP+ astrocytes was detected in the hippocampus of both WT and MPS IIIA mice following chronic low poly(I:C) doses. Large-scale-gene expression studies have shown that astrocytes can be driven towards cytotoxic phenotypes under viral infection, which may be detrimental when triggered during sterile neurodegenerative disease, such as MPS IIIA (Sofroniew, 2014), thus explaining subsequent neuronal damage. Indeed, this phenomenon mainly occurred in the suprapyramidal blade of granule cell layer in the hippocampus, an area where astrocytes cover numerous synapses established between granule neurons and afferents from the entorhinal cortex (Karpf et al, 2022) to regulate learning and memory. While astrocytic damage occurred in brain areas mostly involved in learning and memory, exacerbated astrogliosis was detected in areas related to behaviour, such as cortex and amygdala, potentially triggered by extensive microgliosis. While it appears likely that the perturbation of neurons is driven by damaged astrocytes, the underlying cause of this phenomenon primarily occurring in the hippocampus, and not in other brain regions, requires further clarification. The hippocampus is known to exhibit increased susceptibility to stress and aging (Smith, 1996), possibly owing to its specific location and anatomy, which render it susceptible to cerebrospinal fluid flow disturbances (Lee, 2022). Chronic stress-related damage to the hippocampus is also attributed to adrenal glucocorticoids released to counteract inflammation, as they can compromise energy metabolism and heighten neuronal vulnerability to glutamate excitotoxicity (Smith, 1996). Notably, the hippocampus harbours one of the highest concentrations of glucocorticoid receptor-immunoreactive and mRNA-containing cells (Morimoto et al, 1996). The heightened inflammatory response induced by both acute and chronic poly(I:C) challenges may trigger metabolic

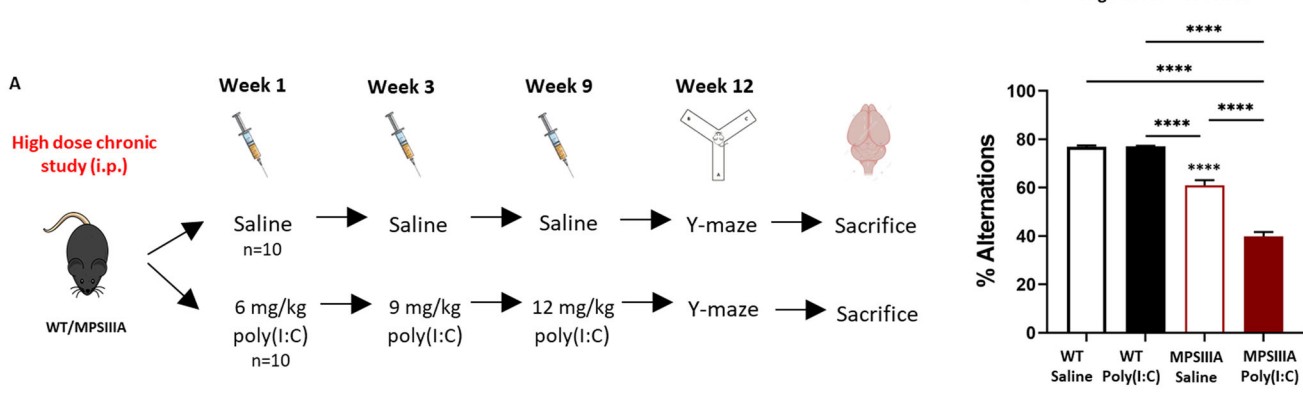

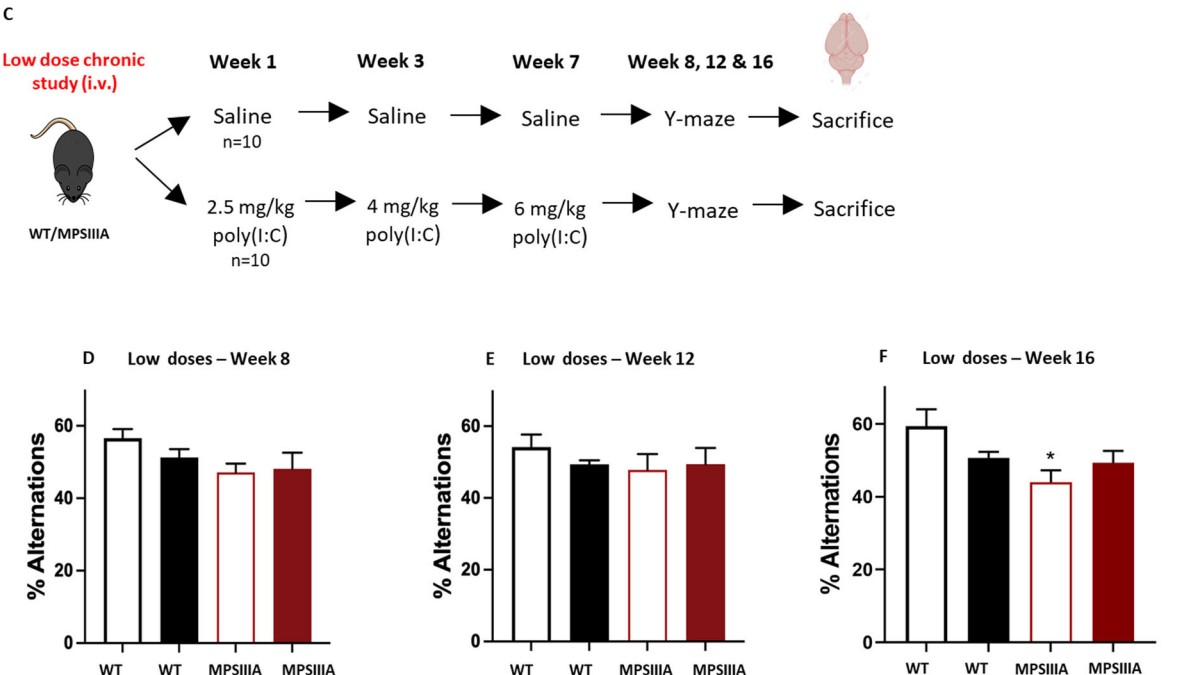

**Figure 5. Chronic administration of poly(I:C) adversely affects MPS IIIA behaviour in a dose-dependent manner.**

(A) Two-month-old WT and MPS IIIA animals were repeatedly challenged with saline or increasing doses (6, 9, 12 mg/kg) of poly(I:C) and sacrificed following completion of behavioural assessment. (B) Cognitive behavioural testing was carried out at 12 weeks from the first injection via Y-maze test ($n = 10$ per group). (C) Two-month-old WT and MPS IIIA animals were repeatedly challenged with saline or increasing doses (2.5, 4, 6 mg/kg) of poly(I:C) and sacrificed following completion of behavioural assessment. Cognitive behavioural testing was carried out after the three low poly(I:C) doses at 8- (D), 12- (E) and 16 weeks (F) from the first injection ($n = 10$–11 per group). Error bars represent standard error of the mean (SEM). Significant differences are determined by two-way ANOVA with Tukey's post hoc analysis. Comparisons showing significant differences are denoted by *$P < 0.05$ and ****$P < 0.0001$ above comparison lines. Significance against WT saline is shown above recorded data. Source data are available online for this figure.

perturbations in the MPS IIIA hippocampus, leading to initial astrocytic damage, as evidenced by a reduction in GFAP positivity. This compromised astrocytic support may result in neuronal damage, as indicated by the loss of NeuN protein, a phenomenon previously observed to decrease following metabolic perturbations (Ünal-Çevik et al, 2004).

Inexplicably, neuronal perturbation did not always correlate with cognitive decline. Notably, stimulation with chronic high poly(I:C) doses led to exacerbation of the pre-existing spatial working memory

deficit in MPS IIIA mice, with no effect on WT animals. Conversely, stimulation with chronic low poly(I:C) doses did not worsen the MPS IIIA behavioural phenotype, suggesting that the effect of systemic immune challenges on behaviour is dose-dependent. There are instances where repeated poly(I:C) dosing can induce tolerance in recipient animals. To avoid this problem, we used increasing dose increments of poly(I:C) in chronic challenges. In addition, previous studies have shown that repeated poly(I:C) challenges do not produce tolerance in behavioural responses (Cunningham et al, 2007), ruling

# Low dose chronic study (i.v.)

**A**

|  | Cortex | Hippocampus | Amygdala |
|---|---|---|---|

WT Saline

WT Poly(I:C)

MPSIIIA Saline

MPSIIIA Poly(I:C)

ILB4

**B** Cortex

**C** Hippocampus

**D** Amygdala

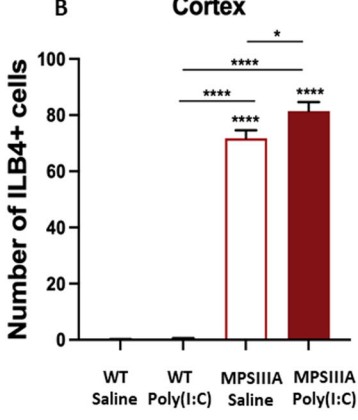

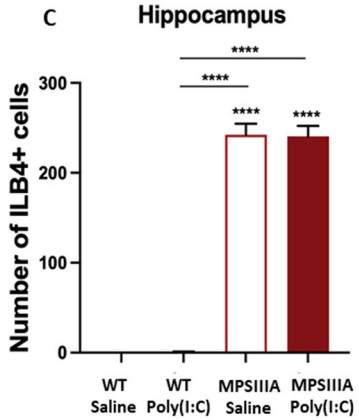

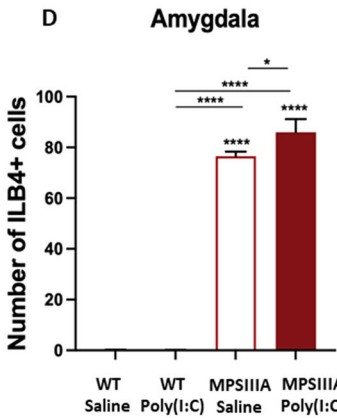

**Figure 6.  Chronic low-dose poly(I:C) administration leads to increased microgliosis in the cortex and amygdala of MPS IIIA mice.**

**(A)** Microglial activation was assessed at 16 weeks after the first poly(I:C) or saline challenge in WT and MPS IIIA mice via isolectin-B4 labelling in the cortex (covering layers IV/V/VI, from 0.26 to −2.70 relative to bregma), hippocampus (−1.70 relative to bregma) and amygdala (−2.70 relative to bregma). Scale bar = 100 µm for cortex and amygdala. Scale bar = 200 µm for hippocampus. Insets demonstrate representative microglia. The number of ILB4+ cells was manually quantified in ImageJ in cortex **(B)**, hippocampus **(C)** and amygdala **(D)**. Error bars represent standard error of the mean (SEM), $n$ = 10–11 in all groups. Significant differences were determined by two-way ANOVA with Tukey's post hoc analysis, *$P$ < 0.05 and ****$P$ < 0.0001. Significance against WT saline is shown above recorded data, all other comparisons are shown with a line. Source data are available online for this figure.

out the possibility that mice might have developed tolerance to low poly(I:C) doses. The confirmation of this lack of tolerance is reinforced in our study through the observed sickness behaviour responses following each poly(I:C) dose. Interestingly, the cognitive impairments that we describe upon chronic high poly(I:C) doses during chronic neurodegeneration have parallels with the cognitive impairments seen in other neurodegenerative and aged animal models (Cunningham et al, 2009; Field et al, 2010). This relationship has also been demonstrated in an Alzheimer's disease patient cohort, where systemic inflammatory events have been correlated with rapid cognitive decline within a 2-month period (Holmes et al, 2003, 2009).

Overall, our data suggest that systemic immune challenges are able to induce a transient inflammatory response in WT mice that does not have a detrimental impact on cognitive function and CNS homeostasis. Conversely, systemic immune challenges potentially exacerbate the already active inflammasome response in a chronic disease such as MPS IIIA. This leads to microglia hyperactivation, characterised by a sustained IL-1β inflammatory response from these cells, potentially causing changes in the astrocytic state and neuronal loss. These data suggest that in neuronopathic lysosomal diseases where inflammasome activation has been described (MPS IIIA, MPSII, Gaucher (Azambuja et al, 2020; Aflaki et al, 2016)— but likely many more), systemic infection can drive neuropathy. As a result, future therapies, whilst rightly focussed on enzyme delivery, should also consider modulation of inflammation either directly or through targeting of inflammasome dependent pathways such as IL-1 (Parker et al, 2020).

## Methods

### Maintenance of mouse colonies

All animal work was approved by the Ethical Review Process Committee of the University of Manchester and in accordance with the Animals (Scientific Procedures) Act, 1986 (UK), under project licence PPL P0C3AEEB0. Mice were housed in groups of 2–5 in individually ventilated cages, with a regular 12/12-h light/dark cycle (7 AM–7 PM), under controlled temperature and light. They were provided with ad libitum access to food and water. MPS IIIA mice on a mixed C57BL/6 J background (B6.Cg-Sgsh^mps3a/6J, Jackson Laboratories, Bar Harbour, MA) were maintained by heterozygote breeding, generating WT and MPS IIIA littermates, with genotyping as previously described (Langford-Smith et al, 2012; Sergijenko et al, 2013).

### Experimental design

Two experimental groups were compared against WT or MPS IIIA control mice. Only female mice were used in our study, as only

MPS IIIA female mice had been reported to recapitulate the behavioural abnormalities observed in patients (Langford-Smith et al, 2011). Male mice from this genotype have a tendency to fight, which often leads to single housing (and changes in their natural behaviour). Histology, biochemistry and behavioural analyses were carried out on $n$ = 6–11. $N$ numbers were based on previous power calculations (Langford-Smith et al, 2012; Sergijenko et al, 2013; Gleitz et al, 2018).

### Poly(I:C) preparation

High molecular weight polyinosinic–polycytidylic acid sodium salt (poly(I:C), tIrl-pic, InvivoGen, Toulouse, France) was prepared for injection by resuspension in sterile NaCl 0.9% (saline) at 1 mg/ml. The solution was then mixed, heated for 10 min at 70 °C to ensure complete solubilisation and then allowed to cool naturally at room temperature for 1 h to ensure proper annealing of double-stranded RNA.

### High-dose intraperitoneal challenge

For acute high dosing, 6 cohorts of 4 month-old female WT or MPS IIIA mice ($n$ = 6 per group) were injected intraperitoneally with either 12 mg/kg of poly(I:C) in 200 µl saline (Field et al, 2010) or saline as a control. Cohorts of 6 mice for poly(I:C) were measured for body weight and sacrificed at 3, 6 and 24 h post injection by an intra-cardiac perfusion of PBS under terminal anaesthesia. The three saline controls were combined ($n$ = 18). The burrowing behaviour was assessed between 5 and 7 h post stimulation in the 6- and 24 h cohorts.

For chronic high dosing, 4 cohorts of 2-month-old female WT or MPS IIIA mice were treated with either three injections of saline or three increasing doses of poly(I:C); 6 mg/kg (week 0; 2 months of age), 9 mg/kg (week 3) and 12 mg/kg (week 9) over a 9-week period (n = 10 per group). At 3, 6 and 24 h post challenge, the body weight was measured. The burrowing behaviour was assessed between 5- and 7-h post stimulation. For both acute and chronic studies, blood was collected via cardiac puncture, allowed to clot at room temperature, centrifuged at 300×$g$ for 10 min, and serum stored at −80 °C. At the end of the chronic study, anesthetised animals received an intra-cardiac perfusion of PBS and the tissues were harvested 12 weeks after the first injection (Gleitz et al, 2018).

### Low-dose intravenous challenge

For acute low dosing, 2- to 4-month-old female WT and MPS IIIA mice ($n$ = 6 per group) were intravenously injected with either a single dose of 6 mg/kg poly(I:C) in 200 µl saline or saline as a control (Homan et al, 1972). At 3, 6 and 24 h post challenge, the

# Low dose chronic study (i.v)

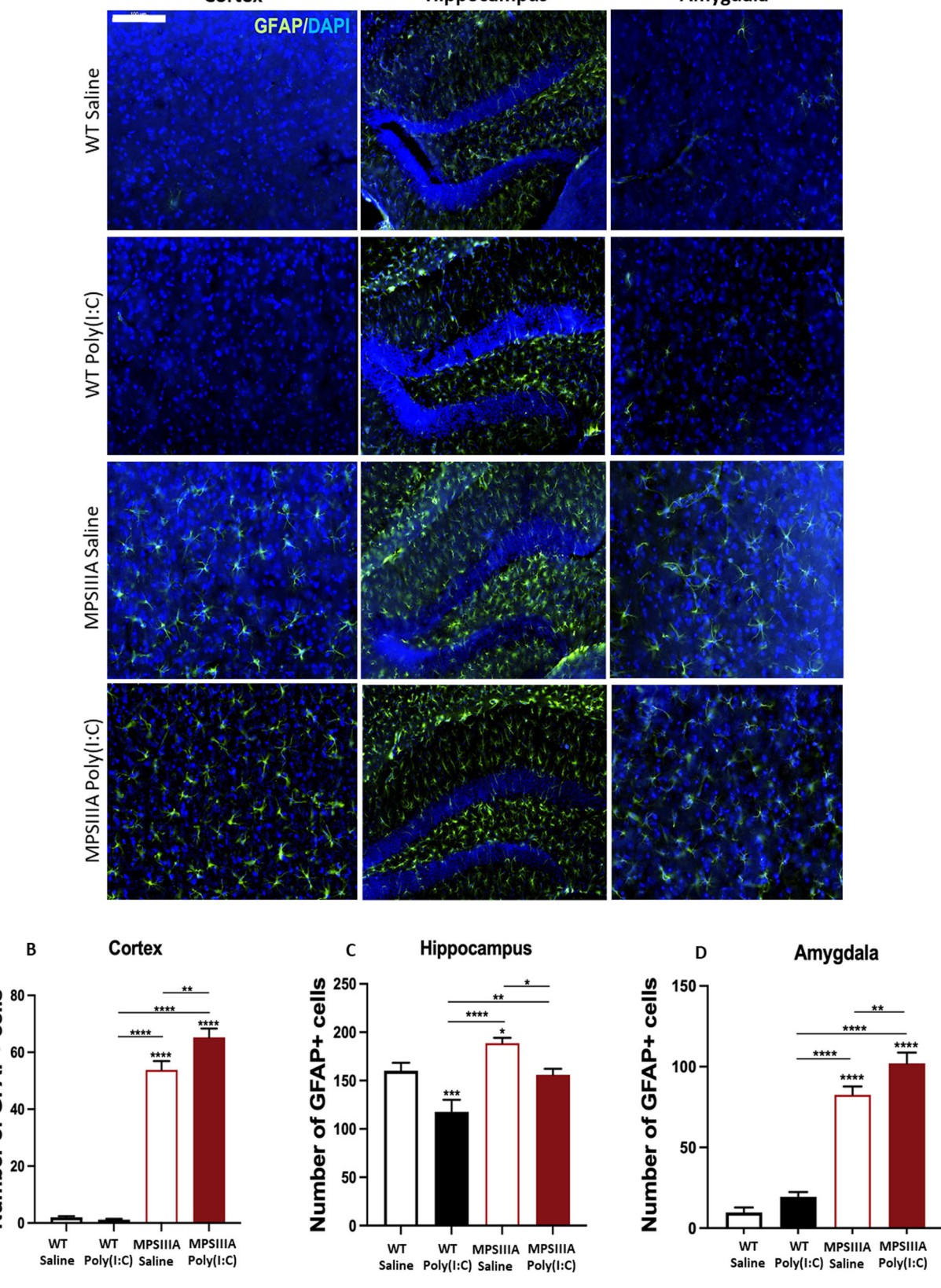

**Figure 7. Chronic low-dose poly(I:C) administration exacerbates astrogliosis in the cortex and amygdala, but leads to GFAP+ astrocyte reduction in the hippocampus of MPS IIIA mice.**

(A) Astrocytic activation was assessed at 16 weeks after the first poly(I:C) or saline challenge in WT and MPS IIIA mice via GFAP labelling in the cortex (covering layers IV/V/VI, from 0.26 to −2.70 relative to bregma), hippocampus (−1.70 relative to bregma) and amygdala (−2.70 relative to bregma). Scale bar = 100 μm for cortex and amygdala. Scale bar = 200 μm for hippocampus. The number of GFAP+ cells was manually quantified in ImageJ in cortex (B), hippocampus (C) and amygdala (D). Error bars represent standard error of the mean (SEM), $n = 10$–11 in all groups. Significant differences were determined by two-way ANOVA with Tukey's post hoc analysis, *$P < 0.05$, **$P < 0.01$, ***$P < 0.001$ and ****$P < 0.0001$. Significance against WT saline is shown above recorded data, all other comparisons are shown with a line. Source data are available online for this figure.

body weight was measured, while the burrowing behaviour was assessed between 5 and 7 h post stimulation. Anesthetised animals received an intra-cardiac perfusion of PBS and the tissues were harvested 24 h after the injection. For chronic high dosing, 2 month old female WT and MPS IIIA mice were treated with either three injections of saline or three increasing doses of poly(I:C); 2.5 mg/kg (week 0; 2 months of age), 4 mg/kg (week 3) and 6 mg/kg (week 7) over a 16-week period ($n = 10$–11 per group). At 3, 6 and 24 h post challenge, the body weight was measured. The burrowing behaviour was assessed between 1 and 3 h post stimulation. At 24 h post challenge, 100 μl blood was withdrawn from the tail and the plasma analysed for cytokine expression. At the end of the study, anesthetised animals received an intra-cardiac perfusion of PBS and the tissues were harvested 16 weeks after the first injection. The original intent for this study was to have a 12-week duration (in line with the high-dose intraperitoneal study). However, since the behavioural analysis was not significant at 12 weeks, we extended the experiment length by 4 more weeks, to see an effect in our control groups.

## Sickness behaviour: burrowing test

The burrowing test was based on sickness behaviour analysis as previously described (Deacon, 2012). The burrowing test measures the activity level of a mouse (food displaced from a feeding tube by burrowing) in a set time and is an indication of how sick an animal is in response to a viral mimetic challenge. In total, 150 g of normal diet food pellets were placed in grey plastic cylinders (20 cm long, 6.8 cm diameter, sealed at one end), which were in turns located in individual mouse cages. The open end was raised by 3 cm above the floor, preventing non-purposeful displacement of the contents. In all, 1–3 h and 5–7 h post injection, mice were tested for a total of 2 h. After this period, the food remaining in the cylinders was weighed and the amount displaced was calculated.

## Y-maze working memory test

Y-maze tests were performed after the third poly(I:C) challenge in chronic treatment groups, when mice were 4-6 months of age. The spontaneous alternation test was used to assess spatial working memory in both control and treatment groups. Spontaneous alternation was assessed during one continuous 10 min session in a Y-maze consisting of three identical arms as previously described (Gleitz et al, 2017). The behaviour was recorded for 10 min, using Top Scan software (Clever Sys. Inc., USA) and a Sony digital camera (Sony). Animals that performed less than 10 entries were excluded from the analysis.

## Tissue harvesting

One brain hemisphere and half of the spleen were snap-frozen on dry ice for biochemistry, and stored at - 80 °C. The other brain hemisphere and a portion of all the above organs were fixed in 4% PFA for minimum 24 h. All the organs, apart from the brain, were transferred from 4% PFA to 70% ethanol and kept at 4 °C for paraffin embedding and slicing. Brains were transferred into 30% sucrose/2 mM $MgCl_2$/1× PBS solution for 48 h and frozen at −80 °C in foil. When needed for flow cytometry analysis, half spleens were collected in RPMI 1640 medium (R0883, Sigma, Poole, UK) + 2% foetal bovine serum (FBS) and kept on ice.

## ELISA

The levels of mouse IL-1β, KC, MCP-1 and MIP-1α were analysed using commercial enzyme-linked immunosorbent assays (ELISAs) according to the manufacturer's protocols (R&D Systems, Minneapolis, USA).

## Immunohistochemistry

Brain tissue sections (30 μm) were cut on a freezing microtome in optimal cutting temperature compound (KP-CryoCompound, 1620 C, Klinipath, Duiven, Netherlands), after removing the cerebellum and olfactory bulb, and stored at 4 °C in TBS-AF buffer (10× TBS; TBS-A: 100 ml 10× TBS, 900 ml $H_2O$, 0.5 g $NaN_3$; TBS-AF: 700 ml TBS-A, 150 g sucrose, 300 ml ethylene glycol) in round-bottomed 96 well plates. Free-floating immunohistochemistry was performed on sections taken 0.02, −1.58 and −2.70 mm from Bregma (where Bregma is the area of the skull where the sagittal and coronal sutures joining the parietal and frontal bones come together) (Wilkinson et al, 2012). Antigen retrieval was performed prior to IL-1β staining by boiling sections in 10 mM citrate buffer (pH 6) for 10 min prior to blocking in 1× PBS/10% goat serum/0.1% Triton X-100. Brain sections were incubated overnight at 4 °C with primary antibodies: rabbit anti-GFAP (1:1500, Z0334; Dako, Stockport, UK), rabbit anti-IL-1β (1:50, 500-P51: Peprotech, London, UK) and rabbit anti-NeuN (1:1000, ab177487; Abcam, Cambridge, UK) or isolectin-B4 (peroxidase-conjugated isolectin-B4 from Bandeiraea simplicifolia, ILB4, 1:200, L5391; Sigma). Sections were washed three times in PBS and goat anti-rabbit Alexa Fluor® 594 (1:1000, A11012; ThermoFisher Scientific, Altrincham, UK) and donkey anti-rabbit Alexa Fluor® 488 (1:1000, A21206: ThermoFisher Scientific) secondary antibodies added as required. The Vectastain ABC system and DAB substrate (Vector Laboratories, Inc., Newark, USA) were used to detect ILB4 and IL-1β antibody binding. All sections were mounted onto positively charged slides (SuperFrost Plus™ Fisher Scientific, Loughborough, UK) and mounted in DPX medium (Thermofisher Scientific) or

## Low dose chronic study (i.v)

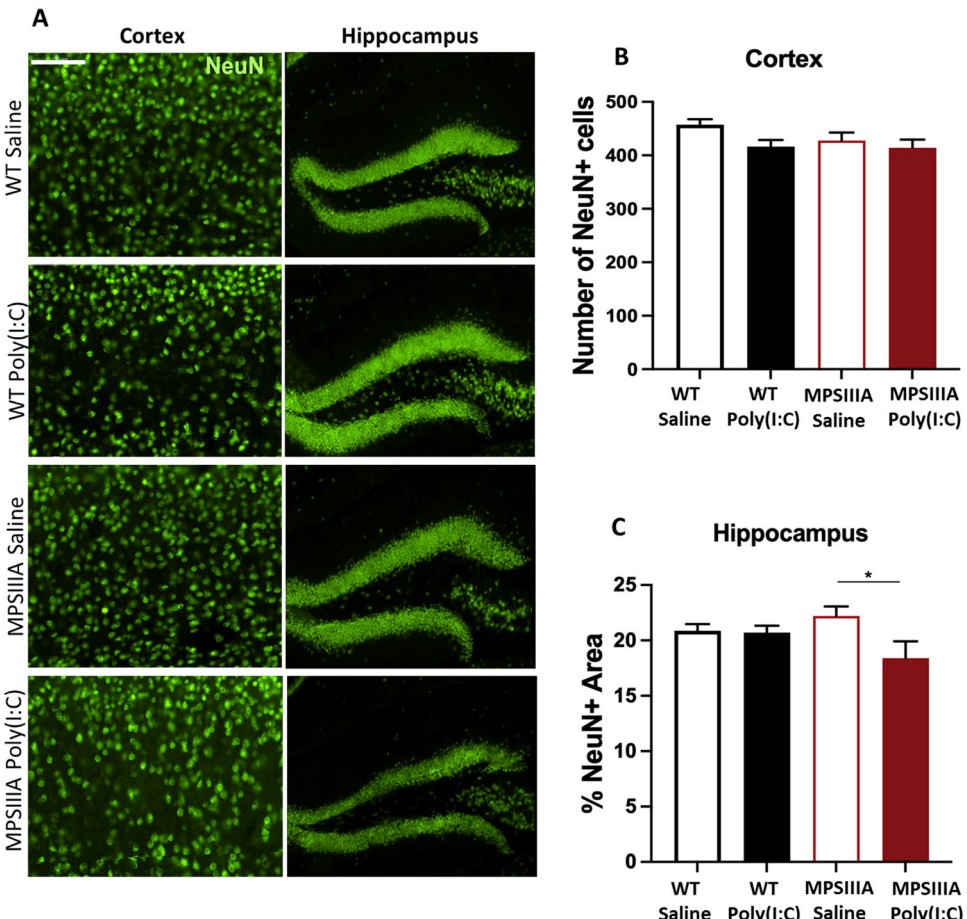

**Figure 8. Chronic low-dose poly(I:C) administration leads to neuronal dropout in the hippocampus of MPS IIIA mice.**

(A) Neuronal dropout was assessed at 16 weeks after the first poly(I:C) or saline challenge in WT and MPS IIIA mice via NeuN labelling in the cortex (covering layers IV/V/VI, from 0.26 to −2.70 relative to bregma) and hippocampus (−1.70 relative to bregma). Scale bar = 100 μm for cortex. Scale bar = 200 μm for hippocampus. (B) The number of NeuN+ cells was quantified in the cortex by CellProfiler. (C) The level of NeuN fluorescence (% NeuN+ area) was quantified in the hippocampus by ImageJ, and normalised against background measurements. Error bars represent standard error of the mean (SEM), $n$ = 10–11 in all groups. Significant differences were determined by two-way ANOVA with Tukey's post hoc analysis, *$P < 0.05$. Source data are available online for this figure.

ProLongTM Gold Antifade mounting medium containing DAPI (Thermofisher Scientific). Images were acquired on a 3D-Histech Pannoramic-250 microscope slide-scanner using a 40x/0.30 Plan Achromat objective (Carl Zeiss) and the DAPI, FITC and TRITC filter sets. Snapshots of the slide-scans were taken using the SlideViewer software (3D-Histech, Budapest, Hungary). GFAP+, ILB4+ and IL-1β+ cells were quantified by ImageJ software. NeuN+ cells were quantified by CellProfiler software. GFAP and NeuN fluorescence were quantified by measuring corrected total cell fluorescence and expressed in arbitrary units.

### qRT-PCR

Total RNA was purified from mouse tissue utilising the TRIzol® reagent RNA isolation procedure (Life Technologies) according to the manufacturer's instructions. Additional DNase treatment was performed by using the turbo DNA-free kit (Invitrogen). The High-Capacity cDNA Reverse Transcription Kit (Applied Biosystems) was used to convert 2 μg total RNA extracted from samples into cDNA, according to the manufacturer's instructions. Quantitative polymerase chain reactions (qPCR) were performed with 2xTaq-Man Universal PCR Master mix (MM), using 20x TaqMan gene expression assays for pro-inflammatory genes (Table EV1) and 20 ng cDNA, according to the manufacturer's instructions. Fold changes in gene expression were calculated as the ratio of molecules of the target gene against the house keeping gene GAPDH (Table EV1), via $\Delta\Delta_{CT}$ analysis.

### Caspase-1 activity assay

Caspase-1 activity was measured as per the manufacturer's protocol (ab39412; Abcam), using recombinant mouse caspase-1 (0.02-2U;

**The paper explained**

**Problem**

Mucopolysaccharidosis IIIA (MPS IIIA) is a rare childhood dementia characterised by progressive brain degeneration. Currently, patients who are ≤2 years have a low chance of neurological benefit following treatment, as brain damage becomes irreversible beyond this age. Notably, MPS IIIA patients seem to deteriorate more sharply following viral infections, making the condition more challenging to treat.

**Results**

The challenge with an acute high dose of the viral mimetic poly(I:C) exacerbated systemic and brain cytokine expression, especially the critical cytokine IL-1β. Following chronic immune challenges with poly(I:C), MPS IIIA mice showed a dose-dependent worsening of cognitive function. Furthermore, these mice exhibited astrocytic and neuronal cell damage in memory-related areas, along with increased glial reactivity in regions associated with behaviour.

**Impact**

Our findings indicate that inflammation contributes to accelerated neurodegeneration in MPS IIIA disease. Consistent with our previous findings, IL-1β emerges as a pivotal catalyst for neuropathological processes in MPS IIIA. Therefore, controlling IL-1β and the inflammasome should be a key focus in developing treatments for neuronopathic lysosomal diseases.

ab52079; Abcam) standards. Plates were incubated at 37 °C for 2 h protected from light, and fluorescence measured at Excitation/Emission = 400/505 nm using a Synergy HT Microplate reader (BioTec, Potton, UK).

## Flow cytometry analysis

The spleen was finely chopped in 1 ml digestion buffer (500 ml H9269-500ML HANKS Sigma, + 0.5% Penicillin-Streptomycin (P/S). Samples were incubated with 1 ml enzyme solution (0.4 U/ml LiberaseTL solution, Sigma 05401020001, + 80U/ml DNAse Deoxyribonuclease I, Sigma D5025-150KU, in digestion buffer) for 15 min at 37 °C. Overall, 100 μl per 1 ml 0.5 M EDTA stop solution (03690-100 ML, Sigma) was added, and the mix was diluted with wash buffer (500 ml DMEM + 0.5% P/S). The digested tissue suspension was poured through a 70-μm cell strainer, the cells were washed through with a further 5 ml of wash buffer and spun at 300×g for 5 min. The pellet was resuspended in 3 ml of red blood cell (RBC) lysis buffer (R7757-100ML, Sigma) and incubated for 3 min at room temperature. Samples were topped up with 5 ml Flow Buffer (PBS + 2% FBS) and spun at 300×g for 5 min. The spleen was resuspended in 5 ml Flow Buffer for cell count. For flow cytometry analysis, $1 \times 10^7$ cells were stained with conjugated antibodies (Table EV2) for 30 min at 4 °C in the dark. Samples were acquired on a BD LSRFortessa™ Flow Cytometer (BD Biosciences), which has the capacity for 7-laser, 18-fluorophore configuration using BD FACSDiva™ software (BD Biosciences). Standardised PMT voltages based on CST were applied prior to acquisition, and compensation was applied to samples using UltraComp eBeads (ThermoFisher Scientific). Flow cytometry data was analysed with FlowJo v10 software (Tree Star). Splenocytes were initially gated by excluding the cells which displayed a size lower than 30k (Fig. EV1), and single cells were identified based on a linear relationship

between FSC-H and FSC-A, followed by SSC-H and SSC-A. Splenocytes were defined as CD45 +, and live cells were selected based on low staining for Live/Dead Zombie UV. Neutrophils were identified within the Lineage+ fraction using CD11b. The CD11b+ $\text{Lin}^{\text{neg}}$ subset was taken forward, and the monocyte population identified as $\text{Ly6C}^{\text{hi}}$ and $\text{MHCII}^{\text{neg}}$. Both macrophages and dendritic cells (DCs) were identified as CD11c+ MHCII +. The DCs were selected from macrophages based on higher expression of both markers. Finally, the two classical DC subsets were identified based on XCR1 (cDC1) and CD11b (cDC2) expression (Fig. EV1).

## Statistical analysis

Animals were recruited on staggered entry, as mice became old enough to be included, thus randomisation was not performed. However, mice were allocated equally across cohorts as recruitment progressed. It was impossible to blind treatment groups in cages, due to the nature of the mouse model and treatments given. Behavioural tests were recorded and analysed at a later time-point from video in a blinded fashion once all tests had been performed. Histochemical analyses were also carried out in a blinded fashion. Statistical analysis was performed using GraphPad Prism 9 software (La Jolla, CA, USA). Two-way analysis of variance (ANOVAs) were performed for multi-group analysis, followed by Tukey's multiple comparison test. Significance was set at $P < 0.05\%$. The number of mice is stipulated by "$n$" in the respective figure legend. Data are presented as standard error of the mean (SEM) unless otherwise stated. Normality testing was performed and those not meeting these criteria were log-transformed to normalise the dataset.

## Data availability

This study includes no data deposited in external repositories.

The source data of this paper are collected in the following database record: biostudies:S-SCDT-10_1038-S44321-024-00092-4.

## Peer review information

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

## Acknowledgements

The Bioimaging Facility Systems Microscopy Centre microscopes used in this study were purchased with grants from BBSRC, Wellcome Trust and the University of Manchester Strategic Fund. Special thanks go to Dr. Roger Meadows for his help with the microscopy. The authors thank the staff of the Manchester Biological Services Facility for assistance. The authors would like to acknowledge the University of Manchester Flow Cytometry Facility. The project was funded by a pilot grant MDBR-22-115-MPS from the Orphan Disease Center at the University of Pennsylvania. A portion of this work was supported by a PhD studentship (HP) awarded by the Neuroscience Research Institute, University of Manchester. BWB is part funded by a grant from the BBSRC BB/X002403/1. For the purpose of open access, the author has applied a Creative Commons Attribution (CCBY) licence to any Author accepted manuscript version arising from this submission.

## Author contributions

**Oriana Mandolfo**: Resources; Data curation; Software; Formal analysis; Validation; Investigation; Visualisation; Methodology; Writing—original draft; Writing—review and editing. **Helen Parker**: Conceptualisation; Resources; Data curation; Software; Formal analysis; Validation; Investigation; Visualisation; Methodology; Writing—review and editing. **Èlia Aguado**: Investigation. **Yuko Ishikawa Learmonth**: Investigation. **Ai Yin Liao**: Methodology. **Claire O'Leary**: Methodology. **Stuart Ellison**: Methodology. **Gabriella Forte**: Methodology. **Jessica Taylor**: Methodology. **Shaun Wood**: Methodology. **Rachel Searle**: Resources. **Rebecca J Holley**: Methodology. **Hervè Boutin**: Conceptualisation; Supervision; Project administration; Writing—review and editing. **Brian W Bigger**: Conceptualisation; Resources; Supervision; Funding acquisition; Project administration; Writing—review and editing.

Source data underlying figure panels in this paper may have individual authorship assigned. Where available, figure panel/source data authorship is listed in the following database record: biostudies:S-SCDT-10_1038-S44321-024-00092-4.

## Disclosure and competing interests statement

BWB is a recipient of licence fees, and unrestricted clinical trial grants from Orchard Therapeutics and AVROBIO for haematopoietic stem cell gene therapies in MPS IIIA and MPSII, respectively, unrelated to this paper. The authors declare no competing interests.

# Expanded View Figures

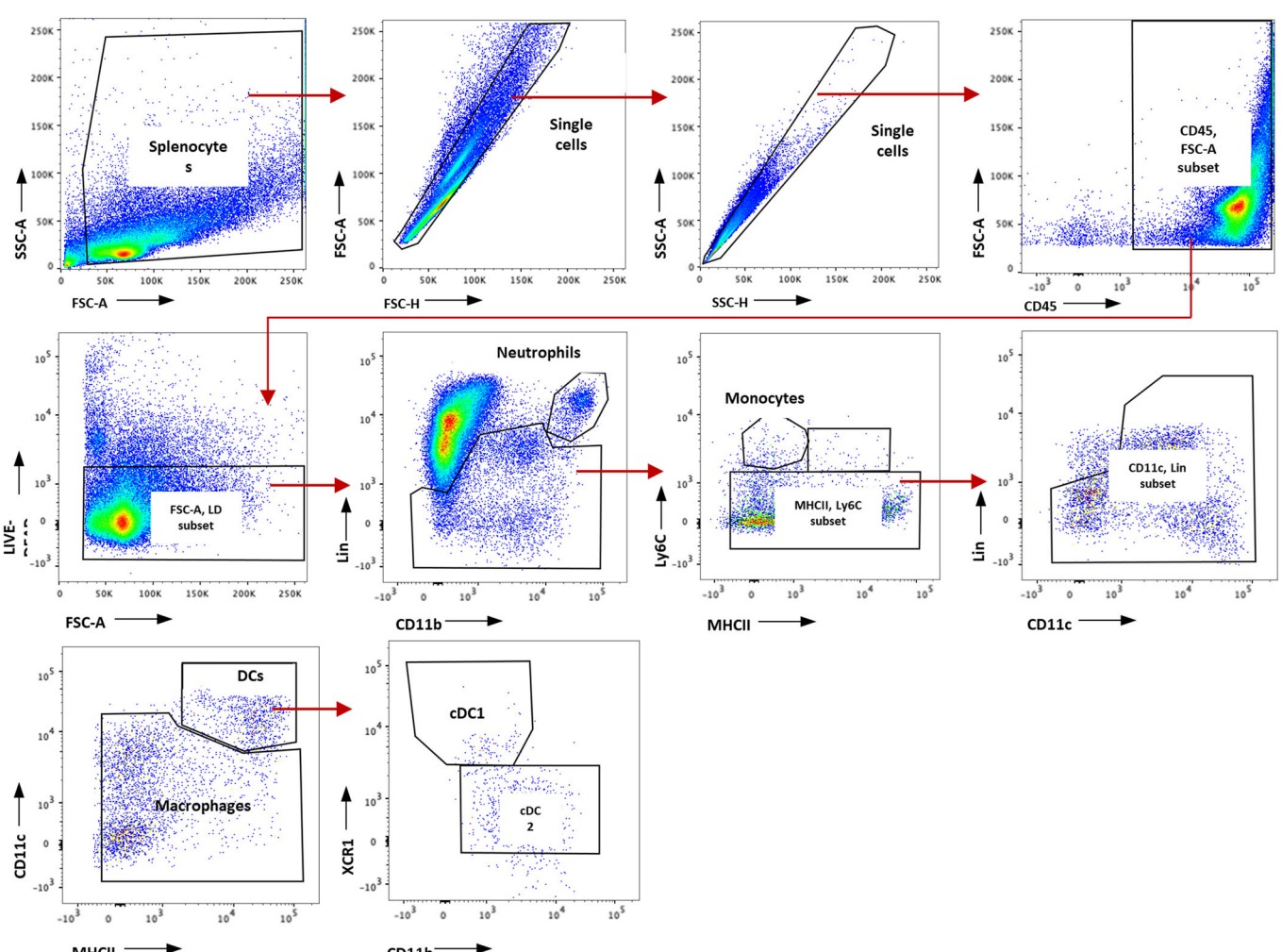

**Figure EV1. Gating strategy for innate immune cells.**

Whole tissue digests were prepared following tissue harvest. Cells were stained with CD45, Live/Dead, Lineage, CD11b, MHCII, Ly6C, CD11c and XRC1. Lymphocytes were initially gated on a forward scatter/side scatter by excluding the cells which displayed a size lower than 30k and were identified based on a linear relationship between FSC-H and FSC-A, followed by SSC-H and SSC-A. Leukocytes were defined as CD45 +, and live leukocytes were selected based on low staining for Live/Dead Zombie UV. From this innate immune cells were selected as follows: neutrophils CD45 +/CD11b +/Lin +, monocytes CD45 +/CD11b +/Lin +/Ly6C +/MHCII +, macrophages and DCs CD45 +/CD11b +/Lin +/Ly6C +/MHCII +/CD11c +.

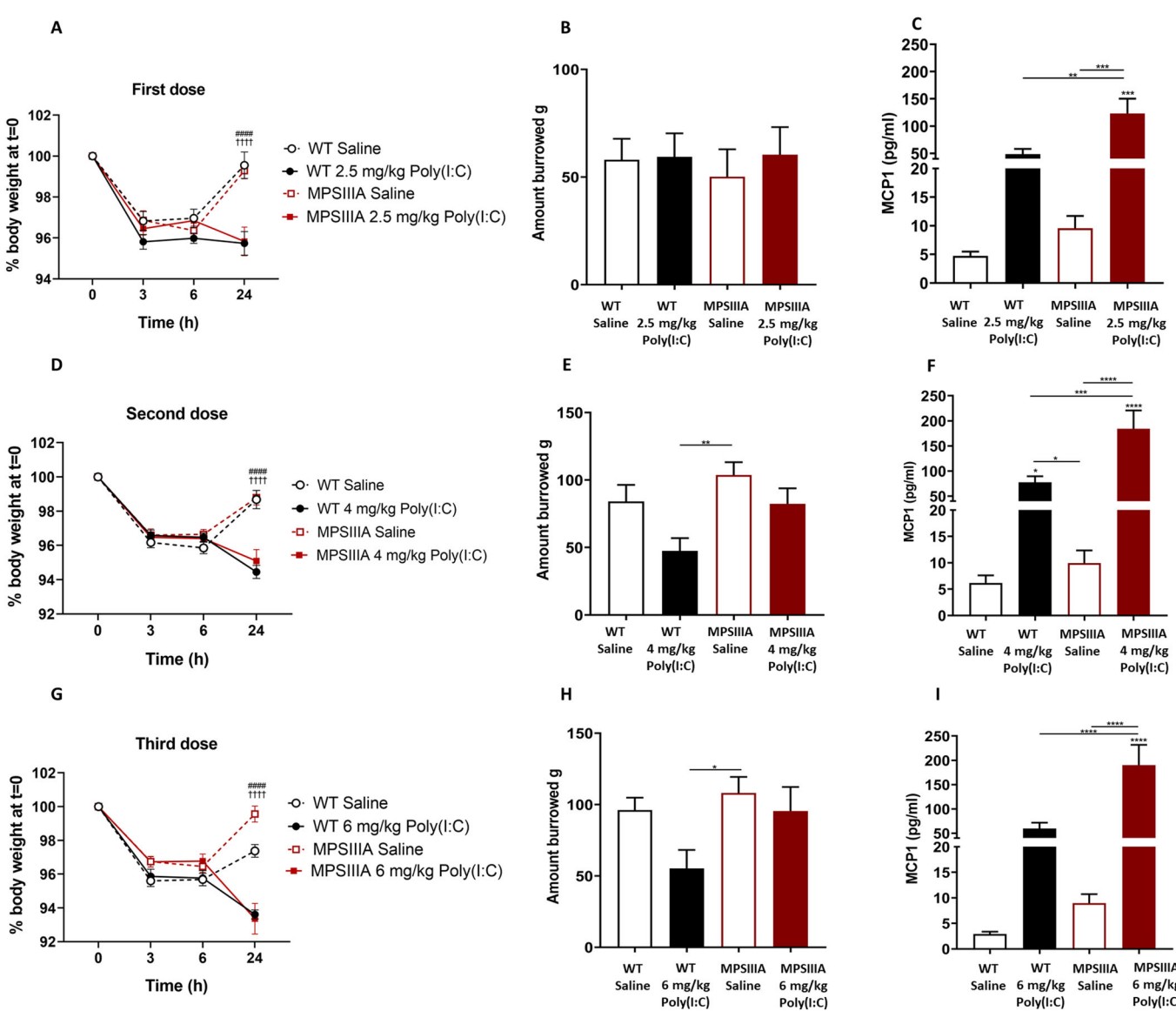

**Figure EV2. Chronic administration of low poly(I:C) doses induces a sickness behaviour response.**

(A) Body weight was measured at the time of poly(I:C) or saline administration ($t = 0$) and again at 3, 6 or 24 h post 2.5 mg/kg poly(I:C) or saline. Body weight is presented as a % of body weight at $t = 0$. (B) Burrowing was assessed between 1–3 h, post 2.5 mg/kg poly(I:C) or saline challenge in WT and MPS IIIA animals. (C) Levels of MPC1 chemokine were analysed in plasma collected at 24 h after challenge with saline or 2.5 mg/kg, poly(I:C).Levels of MCP-1 were evaluated via ELISA ($n = 10$–11). (D) Body weight was measured at the time of poly(I:C) or saline administration ($t = 0$) and again at 3, 6 or 24 h post 4 mg/kg poly(I:C) or saline. Body weight is presented as a % of body weight at $t = 0$. (E) Burrowing was assessed between 1 and 3 h, post 4 mg/kg poly(I:C) or saline challenge in WT and MPS IIIA animals. (F) Levels of MPC1 chemokine were analysed in plasma collected at 24 h after challenge with saline or 4 mg/kg, poly(I:C). Levels of MCP-1 were evaluated via ELISA ($n = 10$–11). (G) Body weight was measured at the time of poly(I:C) or saline administration ($t = 0$) and again at 3, 6 or 24 h 6 mg/kg poly(I:C) or saline. Body weight is presented as a % of body weight at $t = 0$. (H) Burrowing was assessed between 1–3 h, post 6 mg/kg poly(I:C) or saline challenge in WT and MPS IIIA animals. (I) Levels of MPC1 chemokine were analysed in plasma collected at 24 h after challenge with saline or 2.5 mg/kg, poly(I:C).Levels of MCP-1 were evaluated via ELISA ($n = 10$–11). Error bars represent standard error of the mean (SEM). Significant differences are determined by two-way ANOVA with Tukey's post hoc analysis. Comparisons showing significant differences are denoted by *$P < 0.05$, **$P < 0.01$, ***$P < 0.001$ and ****$P < 0.0001$ above comparison lines; significance against WT saline is shown above recorded data (B, C, E, F, H, I). # indicates poly(I:C) being significantly different from saline for WT animals; † indicates poly(I:C) being significantly different from saline for MPS IIIA animals, whereby #### or ††††$P < 0.0001$ above comparison lines (A, D, G).

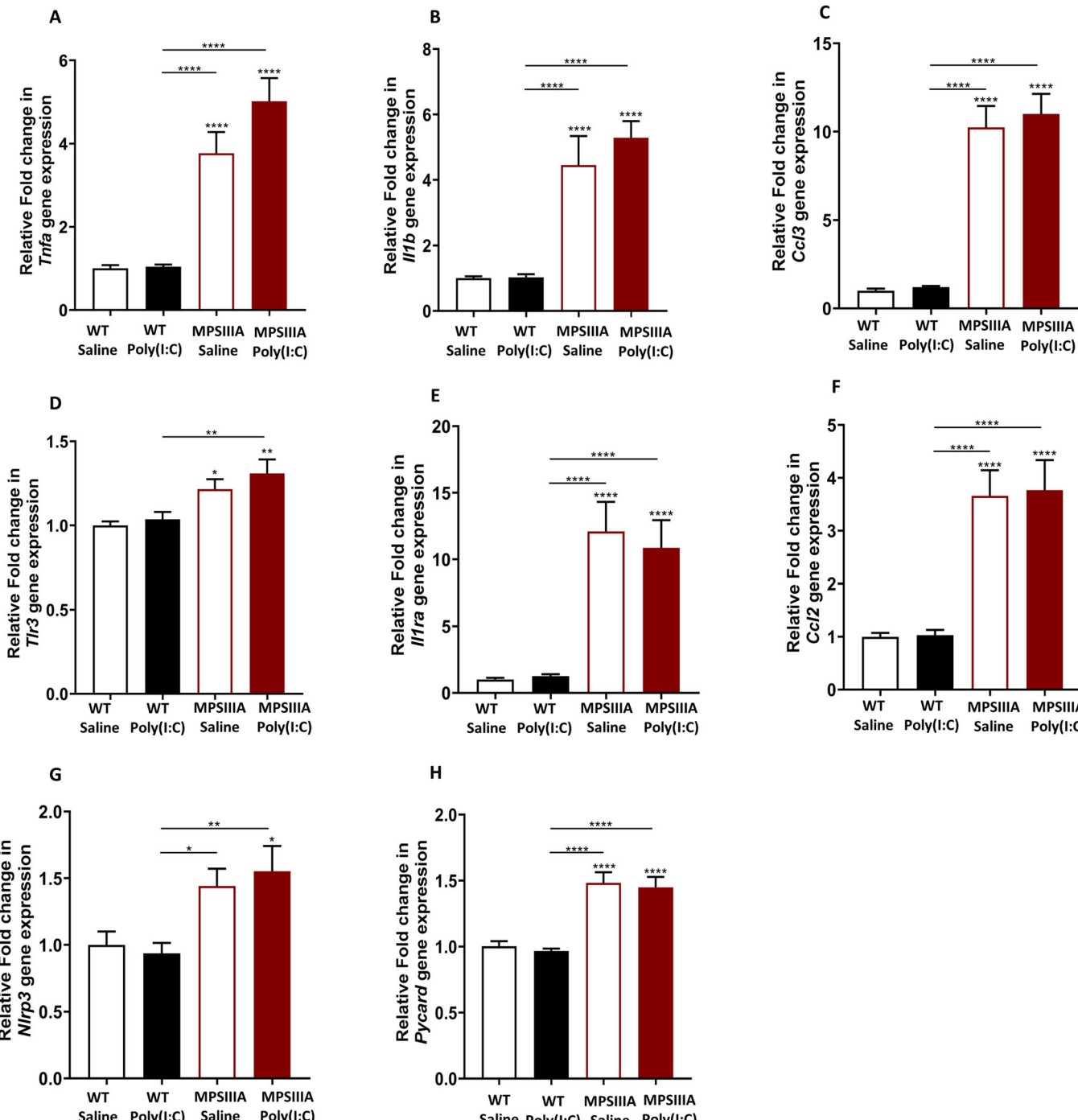

**Figure EV3. Chronic poly(I:C) leads to increased significance of expression of several key pro-inflammatory genes in the CNS of MPS IIIA mice.**

Expression of brain mRNA for *Tnfa* (A), *Il1b* (B), *Ccl3* (C), *Tlr3* (D), *Il1ra* (E), *Ccl2* (F), *Nlrp3* (G), and *Pycard* (H) in WT and MPS IIIA mice either treated with saline or with increasing doses of poly(I:C) ($n = 10$–11 per group). mRNA levels were measured by quantitative PCR and normalised to GAPDH. Error bars represent standard error of the mean (SEM). Significant differences are determined by two-way ANOVA with Tukey's post hoc analysis. Comparisons showing significant differences are denoted by *$P < 0.05$, **$P < 0.01$ and ****$P < 0.0001$ above comparison lines. Significance against WT saline is shown above recorded data.

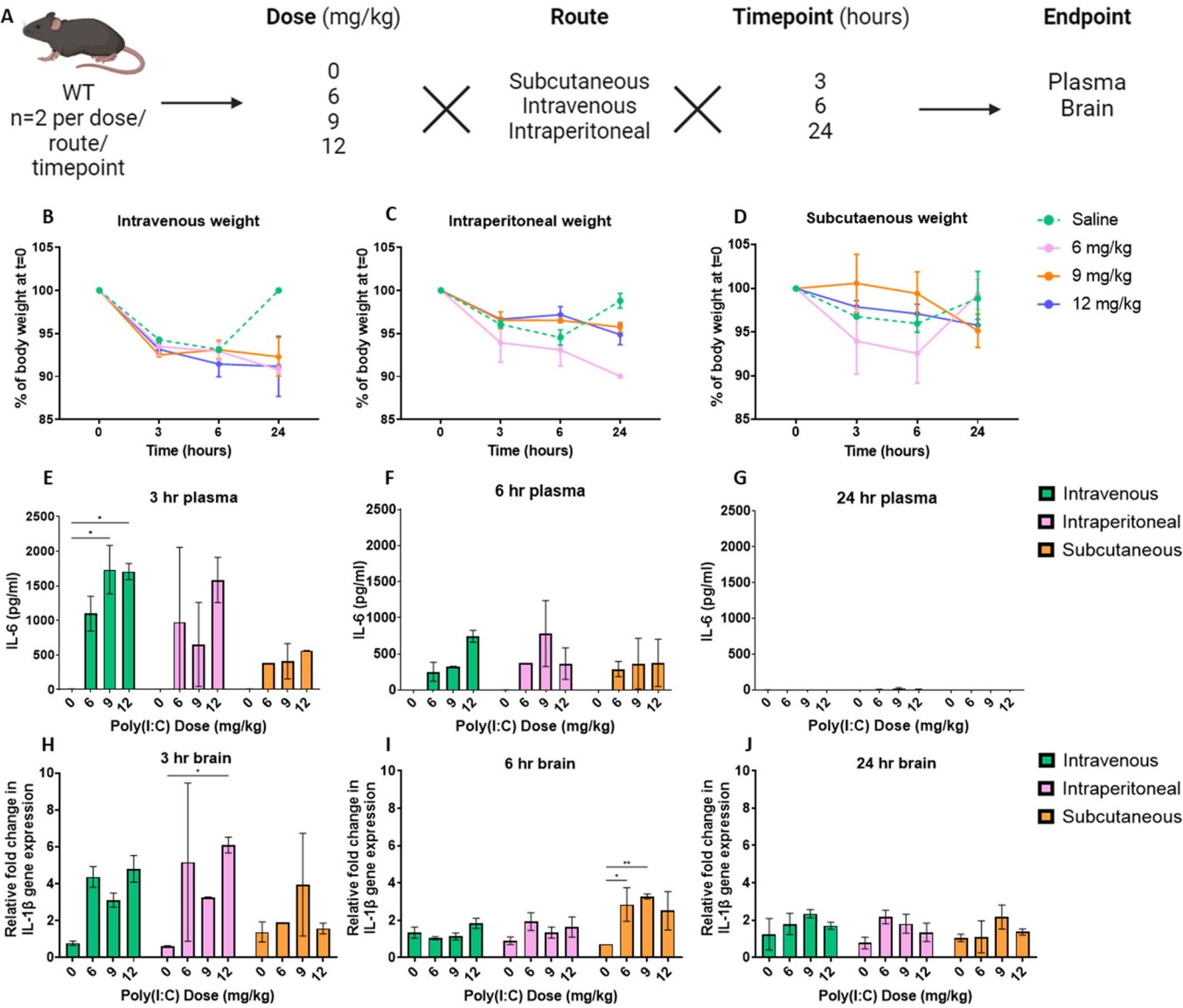

**Figure EV4. Poly(I:C) induces a transient immune response via different delivery routes.**

(A) 8-week-old WT mice were injected with poly(I:C) or saline via subcutaneous, intravenous or intraperitoneal injection. Doses were 6, 9 or 12 mg/kg poly(I:C) or an equivalent volume of saline (n = 2). (B–D) Body weight was measured prior to injection (t = 0), and at 3, 6 and 24 h post injection (n = 2). Data are plotted as mean % body weight at t = 0 and error bars indicate standard deviation (SD). (E–G) Blood was drawn by cardiac puncture at 3, 6 and 24 h post injection. Plasma was separated and tested for IL-6 with ELISA (n = 2). (H–J) Brain tissue was harvested at 3, 6 and 24 h post injection. Protein was isolated by sonication and tested for IL-1β with ELISA. Data are presented as mean and error bars indicate SD. (n = 2). Significant differences were determined by two-way ANOVA with Tukey's post hoc analysis. *$P < 0.05$ and **$P < 0.01$.

