## [Peer Review File · EMBO Molecular Medicine]

Systemic immune challenge exacerbates neurodegeneration in a model of neurological lysosomal disease

Oriana Mandolfo, Helen Parker, Elia Aguado, Yuko Ishikawa Learmonth, Ai Yin Liao, Claire O'Leary, Stuart Ellison, Gabriella Forte, Jessica Taylor, Shaun Wood, Rachel Searle, Rebecca Holley, Herve Boutin, and Brian Bigger

Corresponding author(s): Brian Bigger (brian.bigger@ed.ac.uk)

Review Timeline:

Submission Date:	7th Jul 23
Editorial Decision:	2nd Oct 23
Revision Received:	26th Mar 24
Editorial Decision:	29th Apr 24
Revision Received:	30th May 24
Accepted:	4th Jun 24

Editor: Zeljko Durdevic

Transaction Report:

2nd Oct 2023

Dear Prof. Bigger,

Thank you for the submission of your manuscript to EMBO Molecular Medicine. Please accept my apologies for the unusual delay in getting back to you due to the longer time one referee required to complete evaluation of your manuscript. We have now received feedback from the two reviewers who agreed to evaluate your manuscript. As you will see from the reports below, the referees acknowledge the interest of the study but also raise serious concerns that should be addressed in a major revision. If you would like to discuss further the points raised by the referees, I am available to do so via email or video. Let me know if you are interested in this option.

We would welcome the submission of a revised version within three months for further consideration. Please let us know if you require longer to complete the revision.

I look forward to receiving your revised manuscript.

Yours sincerely,

Zeljko Durdevic

We require:

- 1) A .docx formatted version of the manuscript text (including legends for main figures, EV figures and tables). Please make sure that the changes are highlighted to be clearly visible.
- 2) Individual production quality figure files as .eps, .tif, .jpg (one file per figure). For guidance, download the 'Figure Guide PDF': (<https://www.embopress.org/page/journal/17574684/authorguide#figureformat>).
- 3) A .docx formatted letter INCLUDING the reviewers' reports and your detailed point-by-point responses to their comments. As part of the EMBO Press transparent editorial process, the point-by-point response is part of the Review Process File (RPF), which will be published alongside your paper.
- 4) A complete author checklist, which you can download from our author guidelines (<https://www.embopress.org/page/journal/17574684/authorguide#submissionofrevisions>). Please insert information in the checklist that is also reflected in the manuscript. The completed author checklist will also be part of the RPF.
- 5) Please note that all corresponding authors are required to supply an ORCID ID for their name upon submission of a revised manuscript.

6) It is mandatory to include a 'Data Availability' section after the Materials and Methods. Before submitting your revision, primary datasets produced in this study need to be deposited in an appropriate public database, and the accession numbers and database listed under 'Data Availability'. Please remember to provide a reviewer password if the datasets are not yet public (see <https://www.embopress.org/page/journal/17574684/authorguide#dataavailability>).

13) Author contributions: You will be asked to provide CRediT (Contributor Role Taxonomy) terms in the submission system. These replace a narrative author contribution section in the manuscript.

14) A Conflict of Interest statement should be provided in the main text.

Please note: When submitting your revision you will be prompted to enter your funding and payment information. This will allow Wiley to send you a quote for the article processing charge (APC) in case of acceptance. This quote takes into account any reduction or fee waivers that you may be eligible for. Authors do not need to pay any fees before their manuscript is accepted and transferred to the publisher.

EMBO Press participates in many Publish and Read agreements that allow authors to publish Open Access with reduced/no publication charges. Check your eligibility: <https://authorservices.wiley.com/author-resources/Journal-Authors/open-access/affiliation-policies-payments/index.html>

**** Reviewer's comments ****

Referee #1 (Remarks for Author):

Mandolfo et al

In this manuscript, the authors use the murine model of mucopolysaccharidosis type IIIA (MPS IIIA) to understand the mechanisms that may underlie the clinical observations that MPS IIIA children tend to suffer rapid neurocognitive decline following upper respiratory illnesses.

To do so, the authors treated wild-type and MPS IIIA mice with various doses and routes of administration for polyinosinic/cytidylic (poly I:C), an RNA analogue/viral mimetic, to observe various outcomes including

Acute (intraperitoneal dosing x 1)

Doses = 6 mg/kg (low) or 12 mg/kg (high)

- * Weight
- * Burrowing activity
- * Microglia count
- * Neuronal count
- * Inflammatory cytokines
- * Macrophage count in CNS, spleen

Chronic

IP Doses = 6, 9, 12 mg/kg x 5 weeks

IV Doses = 2.5, 4, 6 mg/kg x 16 weeks

- * Memory (via Y maze)
- * Inflammasome markers in CNS
- * Microglia in CNS
- * Astrocytes

The authors did find high dose acute poly I:C impacted weight negatively and burrowing activity negatively more than low dose acute poly I:C - for both WT and MPS IIIA mice (MPS IIIA more than WT). Also, while MPS IIIA mice already had high levels of CNS microglia, treatment did not further increase microglia but reduced neuronal count. This is in contrast to WT mice who had increased microgliosis but no effect on neuronal count. Other markers of inflammation were increased - seems some, WT more than MPS IIIA (IL-1b, MCP1, Ccl2) while others MPS IIIA more than WT (CD86+ cells, caspases).

Chronic dosing affected memory negatively only in high dose IP MPS IIIA mice, which were already cognitively impaired vs WT when given saline only. No memory effects were seen with low dose chronic IV poly I:C. Only MPS IIIA mice had increased microgliosis and astrocytosis following chronic poly I:C - not WT mice. These results suggest that MPS IIIA mice, which already have upregulated CNS inflammatory processes due to HS storage effects, suffer further inflammation, neuronal loss, and memory deficits following challenge with poly I:C - a proxy inciting agent of inflammation.

The reviewer, having observed such occurrences with not just MPS IIIA but many different LSDs and IEMs, do think the work is noteworthy. Because of the importance of the findings, the reviewer requests significant modifications be made to the manuscript.

First, the rationale for the study design needs more explanation.

- * How were the doses of IV poly I:C determined? The discussion states only how 6 and 12 mg/kg IP were chosen. Provide reference(s) and explain.
- * The lack of parallels makes conclusions difficult for the reader. For example, what is the "equivalent IV dose" of 6 or 12 mg/kg IP? Or, put another way, please provide for readers a way to contextualize what "low dose IV" would translate to if IP, and what "high dose IP" would translate to if IV.
- * Why did chronic dosing need to be incremented? It would have been simpler if one dose were given consistently. The reviewer cannot tell if the neuroinflammatory effects of poly I:C were caused by repeated dosing or because of the most recent, highest dose.
- * Why wasn't cognition tested after an acute dose of poly I:C? The reviewer has seen regression in MPS patients from one single infection. In fact, why was a completely different set of assays done for the acute dose IP and the chronic IP and IV dosing groups
- * Please explain overtly what the burrow test measures. It is left to the reader to deduce that increased weight displaced means more activity.
- * The cohorts for the low dose challenge is confusing. Methods state four groups, N = 11 per group. Was there any different treatments administered to the four groups? The methods describe all received increasing 2.5, 4, then 6 mg/kg dosing
- * Why were only female mice chosen for the high dose IP challenge? Please provide an explanation for why only one sex was used.
- * Were male and female mice chosen for the low dose IV challenge? If so, how were the numbers chosen (N = 11 means an unequal number of M and F mice per cohort)
- * Was the method of euthanization truly intracardiac perfusion by PBS? Is that consistent with local IACUC requirements for rodent euthanasia? Also was blood collected by cardiac puncture prior to euthanasia? That is putting the mice through a lot! Please tidy up the animal welfare description so it doesn't read like you are leaving the mice alive to develop hemopericardium !
- * Why was the duration of survival chosen different for the high dose chronic IP (5 weeks) and the low dose chronic IV (16 weeks)? How can the reader be sure that no further pathology would arise if the chronic IP group was allowed to survive 16 weeks?
- * Figure 1, which explains the acute high and acute low dose study, indicates an N of 6 (acute high) and N of 3 (acute low) which does not align with the methods. Please reconcile discrepancy.
- * The figures do not state explicitly how the I:C was administered (IP or IV). Please do so.
- * Figure 3. Why were serum analytes measured for acute high dose but plasma for acute low dose?
- * Figure 5, which explains the chronic study, has mouse numbers which do not align with methods. N=10 were stated in the figure but N=11 in methods.
- * Were high dose chronic mice sacrificed? The figure implies that they were not.
- * Why couldn't blinding be performed on behavioral and at least histopath assessments? The assessor doesn't need to know genotype and treatment status (see experimental design, "It was impossible to blind in vivo treatment groups." Explain.

Minor issues: a large number of grammar and diction errors, and otherwise needs "tighter" verbiage

- * several times, "seems" and "It is believed" are used. Revise that language please
- * "currently, no effective disease modifying therapy is available for MPS IIIA, with many patients unable to enroll in clinical trials due to rapid degenerative nature of the disease," : use of "with many" connecting two ideas implies therapeutic misconception, that enrollment into clinical trial can generate disease modifying therapy. We don't know that... yet!
- * There are several long run on sentences with multiple "ands." Please split those sentences up!
- * "We have previously demonstrated a two-step model..."
- * "Here, we have tested both acute and chronic ..."
- * Suggest "A more severe cognitive decline in MPSIIIA patients following infection has been Anecdotally reported by several paediatricians. "
- * "his was corroborated in MPS IIIA x IL-1R1 -/- mice, lacking the IL1R1 receptor, which, also" ... too many commas and the reviewer thinks authors meant to start sentence with "This"
- * "viral mimetic; polyinosinic:polycytidylic" - perhaps a comma not semicolon
- * For those not well-versed in marine neuro histopathology, please explain what bregma .02, -1.58, and -2.70 mean
- * Figure 2 legend. Should there be a "" in front of $P < 0.05$?

Referee #2 (Comments on Novelty/Model System for Author):

These aspects can currently only be performed in a suitable animal model, which has been used here.

Referee #2 (Remarks for Author):

This manuscript describes the relationship between artificially induced inflammation caused by viral infection and the effects of MPSIIIA in the central nervous system. The manuscript is interesting as it demonstrates that brain pathology and function may indeed be affected by viral infection. The work also raises several questions as outlined below.

General remark:

- Poly(I:C) is used to mimic viral infection. The authors use a low and high dose. It would be good if the authors could provide information on how this dosing relates to the typical viral load observed during infection. In other words: how well does treatment with poly(I:C) mimic viral infection at the dosing applied? The fact that it is widely used in the literature does not answer this question, the concentrations of double stranded RNA do, as well as the typical and specific responses to double stranded RNA and the absence/presence of responses that only occur after poly(I:C) treatment but not during a typical viral infection.

Main concerns:

- Abstract: The authors state that acute high dose treatment amplifies IL-1 β and caspase 1 expression in the brain of MPS IIIA mice. However, this statement for IL-1 β is based only on staining (not quantitative) performed in hippocampus (not in the entire brain) (Figure 3C). More comprehensive quantification would be informative to sustain the claim of the authors. In particular, the authors demonstrate their capability to measure Il1b gene expression or IL-1 β protein levels at low dose, but they have not conducted these measurements after the acute high dose treatment.
- The authors state that following acute high dose treatment, the increased systemic and brain expression of cytokines results in astrocytic and neuronal loss in hippocampus (fig. 2), and continue stating "Similar levels of cell death, together with exacerbation of gliosis, were also observed in MPS IIIA mice following low chronic poly(I:C) dosing." Yet, the presented data show a decline in NeuN and GFAP markers without demonstrating cell death. Authors should reconsider their conclusion or should demonstrate/confirm cell death. To elaborate on this further: do the authors think that a 50% neuronal loss after 24 hr is indeed the case? If so, saline injection would give an increase of 50% in neurons? This is impossible. More likely, the treatments affect gene expression, and/or other variables play a role in quantification of gene expression as performed in this study. Please reconsider carefully.
- Authors also state that their "data suggests IL-1 β as a key driver of neuropathology in MPS IIIA, thus controlling IL-1 β should be an important component to treat neuronopathic lysosomal diseases." However, while they've established that specific treatment regimens increase this marker in the brain, they haven't mechanistically proven that IL-1 β has a causal role in neuropathology, nor its specificity to the MPS IIIA state compared to WT as for many readouts, including IL-1 β (Figure 3A, 3E), the treatment has an effect on both WT and MPS IIIA animals. Therefore, this conclusion should be modified.

Other concerns:

- Figure 1 burrowing test. The authors mention that n = 3 or 6. While a sample size of 6 may be sufficient for certain behavioral tests and the results presented seem logical, 3 is on the lower side for conducting behavioral tests. To enhance clarity and help the interpretation for other readers, it would be preferable to present the standard deviation (SD) instead of SEM and/or to display individual data points. Alternatively, to increase the n for the low dose.
- Figure 1: The authors claim " This data demonstrates that poly(I:C) induces a dose-dependent sickness behaviour in WT and MPS IIIA mice, which persists past 24 hours following administration." However, the dose dependency is questionable given the low n for the low dose experiment.
- In Figure 2, the authors have provided a detailed description of the acute high dose treatment effects on the hippocampus in both WT and MPS IIIA animals. What justifies the focus on the hippocampus? Considering the results presented in Figure 6, where the effects of a chronic low dose treatment are evident in the cortex and amygdala but not in the hippocampus, it would be beneficial if the authors could also elaborate on the high dose treatment's impact in those regions.
- In Figure 3, the authors show a correlation between the histological (IL-1 β) and biochemical (caspase 1 activity) observations of IL-1 β and the levels of caspase 1 activity following an acute high dose injection. Yet, it is notable that this assay wasn't conducted for the acute low dose experiment within the same figure. What justifies this choice?
- In Figure 2b, the authors present the fold change in fluorescence relative to WT saline for the GFAP marker, whereas in Figure 7, they depict the number of GFAP+ cells. Similarly, Figure 2c displays the fold change of NeuN+ cells, but in Figure 8, the authors provide the percentage of NeuN fluorescence area (hippocampus) or the number of NeuN+ cells (cortex). This inconsistency leads to confusion. The authors should either consistently use one parameter across figures or provide all relevant parameters in each analysis for clarity or justify why they use different parameters.
- Why authors choose to measure levels of MCP1 only in the low dose acute experiment and TNF α only in the high dose? It would be informative for the reader to justify this choice.
- The authors state, "Overall, all the groups treated with poly(I:C) displayed an increase in MCP1 expression in plasma and brain (Figure 3f, h) with the MPS IIIA mice displaying significantly higher response when compared to the WT animals treated with the same poly(I:C) dose." However, based on the data in both Figure 3F and 3H, I don't observe a significantly elevated MCP1 response in MPS IIIA relative to WT after poly(I:C) dose, neither in protein levels nor gene expression. The authors should reconsider their conclusions and adjust the text accordingly.
- The authors state, "Strikingly, a more exacerbated response was observed with regards to monocytes and macrophages in the poly(I:C) treated MPS IIIA mice, when compared to poly(I:C) treated WTs." This statement arises from an n=3 sample size, and the comparisons between treated WT and MPSIIA mice are statistically not different.

Minor points:

- Consistency is needed in the font style to refer to figures in the text: sometimes they are in italic, and at other times they are

not. Please ensure uniformity.

- there are many typos, e.g. "transcriptionand" and "his was corroborated." , among more. Please adjust.
- Ensure a consistent font is used in figures. For instance, Figure 3 A-D uses a different font compared to 3 E-H.
- The authors mention, " When WT mice were treated with low dose poly(I:C), their burrowing activity was significantly reduced by 49% (Figure 1f)." The term "significantly" should be either removed from the statement or its significance should be clearly shown in the figure. Based on the error bars which represent the SEM (and not SD), the results do not seem statistically significant. The authors further state, " Similarly, low dose poly(I:C) treated MPS IIIA mice also displayed a 36% reduction when compared to their saline control (Figure 1f)." The authors should either label this observation as a trend or explicitly show its statistical significance.
- The authors state, " Challenge with an acute high poly(I:C) dose led to a significant microglial activation in the hippocampus of WT animals 3 hours post-poly(I:C) challenge, but no further exacerbation of the existing microgliosis in MPS IIIA animals (Figure 2a)." I do not see the claimed significant microglial activation in the WT mice in the picture provided, it actually looks like a higher background at the 6 hr time point. It might be beneficial to include more representative pictures.
- In Figure 3, why is the high dose measured in serum while the low dose is assessed in plasma? How do serum and plasma measurements compare in this context?
- Please change the sentence on Supplemental Information which can be found on Brain online.

Response to reviewers' comments

REFEREE 1

How were the doses of IV poly I:C determined? The discussion states only how 6 and 12 mg/kg IP were chosen. Provide reference(s) and explain.

We have added a more extensive explanation in our discussion: "These doses were chosen based on previous dose-response studies,²⁷ in which 2, 6 and 12 mg/kg poly(I:C) induced a dose-responsive sickness behaviour, characterised by decreasing locomotor activity, burrowing and body weight, and hyperthermia. Furthermore, 12 mg/kg poly(I:C) proved to induce significant hypothermia and weight loss at later times that is comparable with that caused by influenza virus at 0.1 of its LD₅₀.²⁸ Additionally, we also performed a further dose-response study, in which WT mice were injected with 6, 9 and 12 mg/kg poly(I:C) via three different routes (IP, IV and subcutaneous). Our comparative study indicated a similar sickness behaviour and immunogenic response when administering 12 mg/kg poly(I:C) intravenously, or intraperitoneally and lower responses with 6 mg/kg IV. This is provided in (*Supplementary Figure 4*)."

The lack of parallels makes conclusions difficult for the reader. For example, what is the "equivalent IV dose" of 6 or 12 mg/kg IP? Or, put another way, please provide for readers a way to contextualize what "low dose IV" would translate to if IP, and what "high dose IP" would translate to if IV.

We have provided supplementary data in which we provide a comparison of 6, 9 and 12 mg/kg poly(I:C) doses, administered via IV and IP. 12 mg/kg poly(I:C) delivered IP gives a greater plasma response than 6 mg/kg delivered IV.

Why did chronic dosing need to be incremented? It would have been simpler if one dose were given consistently. The reviewer cannot tell if the neuroinflammatory effects of poly I:C were caused by repeated dosing or because of the most recent, highest dose.

In some instances repeated delivery of poly(I:C) can induce tolerance in recipient animals. As a precaution to avoid this eventuality we used increasing dose increments of poly(I:C) in our chronic challenges. We in fact performed a pilot chronic dosing study, prior to the study presented in this paper, suggesting emergence of tolerance following the second and third dose (where equal doses were given each time). To avoid potential tolerance issues, we incremented the doses in high- and low-dose chronic studies. We have also edited this part in the discussion to clarify the reasons for increasing the doses: "There are instances where repeated poly(I:C) dosing can induce tolerance in recipient animals. To avoid this problem, we used increasing dose increments of poly(I:C) in chronic challenges. Additionally, previous studies have shown that repeated poly(I:C) challenges do not produce tolerance in behavioural responses,²⁷ ruling out the possibility that mice might have developed tolerance to low poly(I:C) doses. The confirmation of this lack of tolerance is reinforced in our study through the observed sickness behaviour responses following each poly(I:C) dose."

Why wasn't cognition tested after an acute dose of poly I:C? The reviewer has seen regression in MPS patients from one single infection. In fact, why was a completely different set of assays done for the acute dose IP and the chronic IP and IV dosing groups.

We thank the reviewer for this point, however we believe we would not be able to show cognitive regression in mice from a single injection as the mouse model has quite a slow onset of behavioural phenotype. We think some of the effect in the chronic dosing is from a sustained lack of brain protection over time. In addition, cognitive decline is only observed in our MPS IIIA mouse model at 5-6 months of age and the original acute dosing was performed at an age where no changes were present in behaviour between WT and MPS IIIA. We have provided additional data to support the direct comparison of the groups.

Please explain overtly what the burrow test measures. It is left to the reader to deduce that increased weight displaced means more activity.

We have edited our methods section to include this explanation: "The burrowing test measures the activity level of a mouse (food displaced from a feeding tube by burrowing) in a set time and is an indication of how sick an animal is in response to a viral mimetic challenge."

The cohorts for the low dose challenge is confusing. Methods state four groups, N = 11 per group. Was there any different treatments administered to the four groups? The methods describe all received increasing 2.5, 4, then 6 mg/kg dosing.

This has been clarified in the methods: "2- to 4-month-old female WT and MPS IIIA mice (n = 6 per group) were intravenously injected with a single dose of 6 mg/kg poly(I:C) or saline in 200 µl saline (acute low dosing). Saline was administered as a control. A second cohort of mice (2-month-old female WT and MPS IIIA mice) were treated three times with either saline or increasing doses of poly(I:C); 2.5 mg/kg, 4 mg/kg and 6 mg/kg over a 16-week period (n = 10 per group) (chronic low dosing)."

Why were only female mice chosen for the high dose IP challenge? Please provide an explanation for why only one sex was used.

We included this explanation in the experimental design section: "Only female mice were used in our study, as only MPS IIIA female mice had been reported to recapitulate the behavioural abnormalities observed in patients (1). Male mice from this genotype have a tendency to fight, which often leads to single housing (and changes their natural behaviour). Thus we stuck to female mice for consistency."

Were male and female mice chosen for the low dose IV challenge? If so, how were the numbers chosen (N = 11 means an unequal number of M and F mice per cohort).

Answered as above and text edited.

Was the method of euthanization truly intracardiac perfusion by PBS? Is that consistent with local IACUC requirements for rodent euthanasia? Also was blood collected by cardiac puncture prior to euthanasia? That is putting the mice through a lot! Please tidy up the animal welfare description so it doesn't read like you are leaving the mice alive to develop hemopericardium !

Rephrased: "anesthetised animals received an intra-cardiac perfusion of PBS and the tissues were harvested". The mice are put under terminal anaesthesia prior to cardiac perfusion with PBS!

Why was the duration of survival chosen different for the high dose chronic IP (5 weeks) and the low dose chronic IV (16 weeks)? How can the reader be sure that no further pathology would arise if the chronic IP group was allowed to survive 16 weeks?

The chronic high dose IP study group was sacrificed at 12 weeks from the start of the experiment (2 months of age at the start – above 5 months of age at the end of the study), with behaviour measured at 12 weeks. The chronic low dose IV study had 3 time points: 8, 12 and 16 weeks since the experiment start. This means that both studies share the 12 week time point and were both destined to have that final time point. However, since our behavioural analysis was not significant at 12 weeks in our low dose chronic IV group, we decided to extend the experiment length by 4 more weeks, in order to see an effect in our control groups.

Figure 1, which explains the acute high and acute low dose study, indicates an N of 6 (acute high) and N of 3 (acute low) which does not align with the methods. Please reconcile discrepancy.

We have now performed a repeat of the acute low dose IV experiment in which 2-month-old female WT and MPS IIIA mice were either treated with saline or 6 mg/kg poly(I:C) (n = 3 per group) to enhance the significance and reliability of our data. Body weight was measured at 0, 3, 6 and 24 hours post-challenge. Burrowing test was performed between 5 and 7 hours post challenge. At 24 hours post-challenge, animals were sacrificed by cardiac perfusion under terminal anaesthesia and blood and brain were harvested. Both figure and methods have now been reconciled and indicate an n=6 for both groups (we have obtained an overall n=6 for the low acute study as our initial and latest low acute study had an n=3).

The figures do not state explicitly how the I:C was administered (IP or IV). Please do so.

Adjusted accordingly. All the figures now are labelled with IP or IV injection.

Figure 3. Why were serum analytes measured for acute high dose but plasma for acute low dose?

We have now performed an extra experiment in order to make our acute studies consistent. We have now performed a repeat of the acute low dose IV experiment in which 2-month-old female WT and MPS IIIA mice were either treated with saline or 6 mg/kg poly(I:C) (n = 3 per group) to enhance the significance and reliability of our data. Body weight was measured at 0, 3, 6 and 24 hours post-challenge. Burrowing test was performed between 5 and 7 hours post challenge. At 24 hours post-challenge, animals were sacrificed by cardiac perfusion under terminal anaesthesia and blood and brain were harvested. This time, serum was isolated from blood and analysed for pro-inflammatory cytokines.

Figure 5, which explains the chronic study, has mouse numbers which do not align with methods. N=10 were stated in the figure but N=11 in methods.

Adjusted accordingly: now both n=10.

Were high dose chronic mice sacrificed? The figure implies that they were not.

Yes! Adjusted accordingly.

Why couldn't blinding be performed on behavioral and at least histopath assessments? The assessor doesn't need to know genotype and treatment status (see experimental design, "It was impossible to blind in vivo treatment groups." Explain.

Rephrased as incorrect: "It was impossible to blind treatment groups in cages, due to the nature of the mouse model and treatments given. Behavioural tests were recorded and analysed at a later time-point from video in a blinded fashion once all tests had been performed. Histochemical analyses were also carried out in a blinded fashion".

Minor issues: a large number of grammar and diction errors, and otherwise needs "tighter" verbiage

- * several times, "seems" and "It is believed" are used. Revise that language please
- * "currently, no effective disease modifying therapy is available for MPS IIIA, with many patients unable to enroll in clinical trials due to rapid degenerative nature of the disease," : use of "with many" connecting two ideas implies therapeutic misconception, that enrollment into clinical trial can generate disease modifying therapy. We don't know that... yet!
- * There are several long run on sentences with multiple "ands." Please split those sentences up!
- * "We have previously demonstrated a two-step model..."
- * "Here, we have tested both acute and chronic ..."
- * Suggest "A more severe cognitive decline in MPSIIIA patients following infection has been Anecdotally reported by several paediatricians. "

* "his was corroborated in MPS IIIA x IL-1R1 -/- mice, lacking the IL1R1 receptor, which, also" ... too many commas and the reviewer thinks authors meant to start sentence with "This"

* "viral mimetic; polyinosinic:polycytidylic" - perhaps a comma not semicolon

* For those not well-versed in marine neuro histopathology, please explain what bregma .02, -1.58, and -2.70 mean

* Figure 2 legend. Should there be a "*" in front of $P < 0.05$?

All corrected accordingly.

REFEREE 2

How well does treatment with poly(I:C) mimic viral infection at the dosing applied? The fact that it is widely used in the literature does not answer this question, the concentrations of double stranded RNA do, as well as the typical and specific responses to double stranded RNA and the absence/presence of responses that only occur after poly(I:C) treatment but not during a typical viral infection.

In vertebrates, double-stranded RNA (dsRNA), a by-product of viral replication, initiates the type I interferon (IFN) response by engaging receptors such as RIG-I, RNA helicase A/DHX9, MDA5, TLR3, and SR-As. Upon detection, dsRNA sets off a signalling cascade involving adaptor proteins, such as IRF3/7, leading to the production of IFN (2). While a real viral infection triggers both innate and adaptive immune responses, involving B cells, T cells, antibody production, and immunological memory, poly(I:C) solely induces an innate immune response. Specifically recognized by endosomal TLR3 and RIG-I-like receptors (RLR) like RIG-I and MDA5, poly(I:C) activates NF κ B and IRF3, resulting in the upregulation of type I interferons (IFN-I), cytokines, and dendritic cell maturation (3).

IFN-I responses induced by both viral infections and poly(I:C) contribute to systemic symptoms observed in acute viral infections, such as fever and reduced locomotor activity (4). To ensure an effect, our poly(I:C) dosing was based on a prior study demonstrating significant hypothermia and weight loss in normal mice following a challenge with 12 mg/kg poly(I:C), comparable to the effects caused by influenza virus at 0.1 of its LD₅₀ (5). We incrementally increased Poly(I:C) dosing in chronic studies to avoid the problem of tolerance, that has occasionally been reported on sequential dosing with the same dose.

We have reworded and added this explanation to our discussion together with the reference comparing poly(I:C) with influenza virus, in order to help the reader better understand this point.

Abstract: The authors state that acute high dose treatment amplifies IL-1 β and caspase 1 expression in the brain of MPS IIIA mice. However, this statement for IL-1 β is based only on staining (not quantitative) performed in hippocampus (not in the entire brain) (Figure 3C). More comprehensive quantification would be informative to sustain the claim of the authors. In particular, the authors demonstrate their capability to measure Il1b gene expression or IL-1 β protein levels at low dose, but they have not conducted this measurements after the acute high dose treatment.

This is a reasonable point. Since the increase in IL-1 β was only observed in the hippocampus, we have adjusted the statement in our abstract accordingly: "Challenge with an acute high poly(I:C) dose exacerbated systemic and brain cytokine expression, particularly IL-1 β in the hippocampus. This was accompanied by an increase in caspase-1 activity within the brain, ultimately culminating in the loss of GFAP+ astrocytes and NeuN+ neurons in the hippocampal region of MPS IIIA mice."

In addition, we have also now added gene expression analysis for IL-1 β , TNF α and CCL2. The data indicates an upregulation in IL-1 β gene expression levels at 3 and 6 hours post-high dose poly(I:C) administration in the MPS IIIA mice and retained upregulation until 24 hours. This further supports the conclusion above.

The authors state that following acute high dose treatment, the increased systemic and brain expression of cytokines results in astrocytic and neuronal loss in hippocampus (fig. 2), and continue stating "Similar levels of cell death, together with exacerbation of gliosis, were also observed in MPS IIIA mice following low chronic poly(I:C) dosing." Yet, the presented data show a decline in NeuN and GFAP markers without demonstrating cell death. Authors should reconsider their conclusion or should demonstrate/confirm cell death. To elaborate on this further: do the authors think that a 50% neuronal loss after 24 hr is indeed the case? If so, saline injection would give an increase of 50% in neurons? This is impossible. More likely, the treatments affect gene expression, and/or other variables play a role in quantification of gene expression as performed in this study. Please reconsider carefully.

This is also a reasonable point. We do not have evidence of cell death per se in the acute setting— just loss of marker expression, which doesn't necessarily mean loss of cells in an acute setting. This response is also very close after poly(I:C) dosing, so losing GFAP expression briefly is certainly possible, without necessarily losing neurons or astrocytes.

We have changed our conclusion to suggest a possible hypothesis for what is occurring:

"While it appears likely that the perturbation of neurons is driven by damaged astrocytes, the underlying cause of this phenomenon primarily occurring in the hippocampus requires further clarification. The hippocampus is known to exhibit increased susceptibility to stress and aging (6), possibly owing to its specific location and anatomy, which render it susceptible to cerebrospinal fluid flow disturbances (7). Chronic stress-related damage to the hippocampus is also attributed to adrenal glucocorticoids released to counteract inflammation, as they can compromise energy metabolism and heighten neuronal vulnerability to glutamate excitotoxicity (6). Notably, the hippocampus harbors one of the highest concentrations of glucocorticoid receptor-immunoreactive and mRNA-containing cells (8). The heightened inflammatory response induced by both acute and chronic poly(I:C) challenges is likely to trigger metabolic perturbations in the MPS IIIA hippocampus, leading to initial astrocytic damage, as evidenced by a reduction in GFAP positivity. This compromised astrocytic support ultimately results in neuronal damage, as indicated by the loss of NeuN protein, a phenomenon previously observed to decrease following metabolic perturbations (9)."

Authors also state that their "data suggests IL-1 β as a key driver of neuropathology in MPS IIIA, thus controlling IL-1 β should be an important component to treat neuronopathic lysosomal diseases." However, while they've established that specific treatment regimens increase this marker in the brain, they haven't mechanistically proven that IL-1 β has a causal role in neuropathology, nor its specificity to the MPS IIIA state compared to WT as for many readouts, including IL-1 β (Figure 3A, 3E), the treatment has an effect on both WT and MPS IIIA animals. Therefore, this conclusion should be modified.

We understand this point and have adjusted this section of the abstract accordingly: "While further investigation is warranted to fully understand the extent of IL-1 β involvement in the exacerbated response observed in MPS IIIA, our data robustly reinforces our previous findings, indicating IL-1 β as a pivotal catalyst for neuropathological processes in MPS IIIA."

In fact, we do have published work demonstrating the IL-1 β role in MPS IIIA neuropathology (10).

Figure 1 burrowing test. The authors mention that $n = 3$ or 6 . While a sample size of 6 may be sufficient for certain behavioural tests and the results presented seem logical, 3 is on the lower side for conducting behavioural tests. To enhance clarity and help the interpretation for other readers, it would be preferable to present the standard deviation (SD) instead of SEM and/or to display individual data points. Alternatively, to increase the n for the low dose.

We have now performed a repeat of the acute low dose IV experiment in which 2-month-old female WT and MPS IIIA mice were either treated with saline or 6 mg/kg poly(I:C) ($n = 3$ per group) to enhance the significance and reliability of our data. Body weight was measured at 0 , 3 , 6 and 24 hours post-challenge. Burrowing test was performed between 5 and 7 hours post challenge. At 24 hours post-challenge, animals were sacrificed by cardiac perfusion under terminal anaesthesia and blood and brain were harvested. We have obtained an overall $n=6$ for the low acute study as our initial and latest low acute study had an $n=3$.

Figure 1: The authors claim " This data demonstrates that poly(I:C) induces a dose-dependent sickness behaviour in WT and MPS IIIA mice, which persists past 24 hours following administration." However, the dose dependency is questionable given the low n for the low dose experiment.

We have now performed an additional low acute poly(I:C) experiment, which brings our total number to $n=6$ (as per explanation above).

In Figure 2, the authors have provided a detailed description of the acute high dose treatment effects on the hippocampus in both WT and MPS IIIA animals. What justifies the focus on the hippocampus? Considering the results presented in Figure 6, where the effects of a chronic low dose treatment are evident in the cortex and amygdala but not in the

hippocampus, it would be beneficial if the authors could also elaborate on the high dose treatment's impact in those regions.

In our acute high dose study, we only observed changes in the hippocampal area, while no changes were observed in other areas, including cortex and amygdala. We have now added a statement in the result section (Figure 2) to clarify this: "Notably, changes in gliosis levels were exclusive to the hippocampus following poly(I:C) treatment." Furthermore, we posit that the majority of the behavioural effects stem from hippocampal activity, as working memory is widely attributed to this brain region. Thus, our emphasis on investigating this specific area.

In Figure 3, the authors show a correlation between the histological (IL-1 β) and biochemical (caspase 1 activity) observations of IL-1 β and the levels of caspase 1 activity following an acute high dose injection. Yet, it is notable that this assay wasn't conducted for the acute low dose experiment within the same figure. What justifies this choice?

We have now performed IHC staining for IL-1 β in the low acute IV study and have not observed a significant increase when comparing the MPS IIIA poly(I:C)-treated group with the saline control. Following this, the caspase-1 activity assay was not performed.

In Figure 2b, the authors present the fold change in fluorescence relative to WT saline for the GFAP marker, whereas in Figure 7, they depict the number of GFAP+ cells. Similarly, Figure 2c displays the fold change of NeuN+ cells, but in Figure 8, the authors provide the percentage of NeuN fluorescence area (hippocampus) or the number of NeuN+ cells (cortex). This inconsistency leads to confusion. The authors should either consistently use one parameter across figures or provide all relevant parameters in each analysis for clarity or justify why they use different parameters.

Our preference is to count the number of NeuN+ and GFAP+ cells, as this gives us a good indication of neuronal degeneration and astrogliosis levels. However, in some instances, especially in the hippocampus, cell density is so high that we had to switch our analysis from cell count to intensity of staining. In previous studies, we have also compared the outcomes relative to cell count and intensity of staining in several areas of the brain and observed that they perform in a similar manner.

Why authors choose to measure levels of MCP1 only in the low dose acute experiment and TNF α only in the high dose? It would be informative for the reader to justify this choice.

We have now made the two studies consistent in terms of output analyses and added these data.

The authors state, "Overall, all the groups treated with poly(I:C) displayed an increase in MCP1 expression in plasma and brain (Figure 3f, h) with the MPS IIIA mice displaying significantly higher response when compared to the WT animals treated with the same poly(I:C) dose." However, based on the data in both Figure 3F and 3H, I don't observe a significantly elevated MCP1 response in MPS IIIA relative to WT after poly(I:C) dose, neither in protein levels nor gene expression. The authors should reconsider their conclusions and adjust the text accordingly.

The data regarding the MCP1 plasma levels in the acute low poly(I:C) study was removed to ensure consistency with the data provided in the acute high poly(I:C) study. We now have an overall n=6 in our low acute dose study (as explained in the previous sections) and this has led to statistical significance in the CCL2 gene expression levels in WT and MPS IIIA poly(I:C)-treated groups when compared to their saline controls. We now state: "Furthermore, a 4- and a 2.2-fold increase in brain *Ccl2* gene expression was also observed in the poly(I:C)-treated WT and MPS IIIA respectively, when compared to their corresponding saline controls (*Figure 3I*)."

The authors state, "Strikingly, a more exacerbated response was observed with regards to monocytes and macrophages in the poly(I:C) treated MPS IIIA mice, when compared to poly(I:C) treated WT mice." This statement arises from an n=3 sample size, and the comparisons between treated WT and MPS IIIA mice are statistically not different.

This has been reworded as follows: "Strikingly, a more exacerbated trend was observed with regard to monocytes and macrophages in the poly(I:C)-treated MPS IIIA mice compared to poly(I:C)-treated WT mice."

Consistency is needed in the font style to refer to figures in the text: sometimes they are in italic, and at other times they are not. Please ensure uniformity.

Adjusted accordingly.

There are many typos, e.g. "transcriptionand" and "his was corroborated." , among more. Please adjust.

Adjusted accordingly.

Ensure a consistent font is used in figures. For instance, Figure 3 A-D uses a different font compared to 3 E-H.

Consistent font is now used.

The authors mention, " When WT mice were treated with low dose poly(I:C), their burrowing activity was significantly reduced by 49% (*Figure 1f*)." The term "significantly" should be either removed from the statement or its significance should be clearly shown in the figure. Based on the error bars which represent the SEM (and not SD), the results do not seem statistically significant. The authors further state, " Similarly, low dose poly(I:C) treated MPS IIIA mice also displayed a 36% reduction when compared to their saline control (*Figure 1f*)." The authors should either label this observation as a trend or explicitly show its statistical significance.

This has been changed accordingly, as significances and values have changed following our additional low acute experiment.

The authors state, " Challenge with an acute high poly(I:C) dose led to a significant microglial activation in the hippocampus of WT animals 3 hours post-poly(I:C) challenge, but no further exacerbation of the existing microgliosis in MPS IIIA animals (*Figure 2a*)." I do not see

the claimed significant microglial activation in the WT mice in the picture provided, it actually looks like a higher background at the 6 hr time point. It might be beneficial to include more representative pictures.

We thank the reviewer for this point. However, the claimed microglial activation in the WT mice at 6 hours is based on the quantification of ILB4+ cells in the hippocampus provided with the graph.

In Figure 3, why is the high dose measured in serum while the low dose is assessed in plasma? How do serum and plasma measurements compare in this context?

We have now conducted all of our analyses on serum to eliminate this discrepancy.

Please change the sentence on Supplemental Information which can be found on Brain online.

Done.

REFERENCES

1. Langford-Smith A, Langford-Smith KJ, Jones SA, Wynn RF, Wraith JE, Wilkinson FL, et al. Female Mucopolysaccharidosis IIIA Mice Exhibit Hyperactivity and a Reduced Sense of Danger in the Open Field Test. *PLoS ONE*. 2011;6(10):e25717.
2. Poynter SJ, Dewitte-Orr SJ. Understanding Viral dsRNA-Mediated Innate Immune Responses at the Cellular Level Using a Rainbow Trout Model. *Frontiers in Immunology*. 2018;9.
3. McGarry N, Murray CL, Garvey S, Wilkinson A, Tortorelli L, Ryan L, et al. Double stranded RNA drives innate immune responses, sickness behavior and cognitive impairment dependent on dsRNA length, IFNAR1 expression and age. 2021.
4. Traynor TR, Majde JA, Bohnet SG, Krueger JM. Intratracheal double-stranded RNA plus interferon- γ : A model for analysis of the acute phase response to respiratory viral infections. *Life sciences*. 2004;74(20):2563-76.
5. Song Y, Wang X, Zhang H, Tang X, Li M, Yao J, et al. Repeated Low-Dose Influenza Virus Infection Causes Severe Disease in Mice: a Model for Vaccine Evaluation. *Journal of Virology*. 2015;89(15):7841-51.
6. Smith MA. Hippocampal vulnerability to stress and aging: possible role of neurotrophic factors. *Behavioural Brain Research*. 1996;78(1):25-36.
7. Lee SJ. Alzheimer's disease is a result of loss of full brain buoyancy. *Medical hypotheses*. 2022;164:110857.
8. Morimoto M, Morita N, Ozawa H, Yokoyama K, Kawata M. Distribution of glucocorticoid receptor immunoreactivity and mRNA in the rat brain: an immunohistochemical and in situ hybridization study. *Neuroscience research*. 1996;26(3):235-69.
9. Ünal-Çevik I, Kılınç M, Gürsoy-Özdemir Y, Gurer G, Dalkara T. Loss of NeuN immunoreactivity after cerebral ischemia does not indicate neuronal cell loss: a cautionary note. *Brain research*. 2004;1015(1-2):169-74.
10. Parker H, Ellison SM, Holley RJ, O'Leary C, Liao A, Asadi J, et al. Haematopoietic stem cell gene therapy with IL-1Ra rescues cognitive loss in mucopolysaccharidosis IIIA. *EMBO Molecular Medicine*. 2020;12(3).

29th Apr 2024

Dear Prof. Bigger,

Thank you for the submission of your revised manuscript to EMBO Molecular Medicine. I am pleased to inform you that we will be able to accept your manuscript pending the following final amendments:

- 1) Please address all referee #1 points and implement all his/her suggestions.
- 2) Authors: We note a discrepancy of author's name: Aiyin Liao in the manuscript file vs. Ai Yin Liao in our submission system. Please correct.
- 3) In the main manuscript file, please do the following:
 - Please address all comments suggested by our data editors listed below:
 1. Please note that a separate 'Data Information' section is required in the legends of figures 1b-c, e-f; 2a-c; 3a-e, g-l, n; 4a-e; 6b-d; 7a-d; EV 2a-i; EV 4e-j.
 2. Please note that the legends for figures 1b-e is not provided in the sequential manner (legend for figure 1d is provided before legends of figures 1b-c, legend for figure 1e is provided before legend of figure 1c). This needs to be rectified.
 3. Please note that the legend for figure 5c is not provided in the manuscript. This needs to be rectified.
 4. Please note that the legend for figure 8c is missing in the manuscript. This needs to be rectified.
 5. Please note that the legends for figures EV 2b-h is not provided in the sequential manner (legends for figures EV 2d, g is provided before legends of figures 2b-c and 2b-f respectively, legend for figure 2e, h is provided before legends of figures 2c and 2c, f, i, respectively). This needs to be rectified.
 6. Please note that in figures 3a-e, g-l, n; 5b, f; 6b-d; EV 3a-h; there is a mismatch between the annotated p values in the figure legend and the annotated p values in the figure file that should be corrected.
 7. Please note that for the figure 8b, p-values and statistical tests are indicated in the legends. However, comparison for the same, ""****/**/*/*"" has not been represented in the figure. Please rectify this in the figure or legend as applicable.
 8. Please note that information related to n is missing in the legends of figures 1b, e; 3n; EV 4b-d.
 9. Please note that n=2 in figures EV 4e-j.
 - Correct order of the manuscript sections as follows: Abstract, Keywords, Introduction, Results, Discussion, Methods, Acknowledgements, Disclosure and competing interests statement, References, Figure legends, Tables and their legends, Expanded View Figure legends.
 - Limit keywords to max 5.
 - Please add callout for Fig 3l.
 - Statistical paragraph should reflect all information that you have filled in the Authors Checklist, especially regarding randomization, blinding, replication etc. Indicate in legends exact p= values, not a range, along with the statistical test used. To keep the figures "clear" some authors found providing an Appendix table Sx with all exact p-values preferable. You are welcome to do this if you want to.
 - Please rename "Conflict of interest" to "Disclosure Statement & Competing Interests". We updated our journal's competing interests policy in January 2022 and request authors to consider both actual and perceived competing interests. Please review the policy <https://www.embopress.org/competing-interests> and update your competing interests if necessary.
 - Please correct the reference citation in the text and reference list. In the text of the manuscript, a reference should be cited by author and year of publication. Include a space between a word and the opening parenthesis of the reference that follows. In the reference list, citations should be listed in alphabetical order. Where there are more than 10 authors on a paper, 10 will be listed, followed by "et al.". Please check "Author Guidelines" for more information. <https://www.embopress.org/page/journal/17574684/authorguide#referencesformat>
 - Please provide data availability statement. If no data were deposited add the sentence "This study includes no data deposited in external repositories".
- 4) Remove ARRIVE Author Checklist.
- 5) Please upload Table EV1 and EV2 as separate files.
- 6) Funding: Please merge it with "Acknowledgement. If the project number is available, please include it.
- 7) The Paper Explained: Please provide "The Paper Explained" and add it to the main manuscript text. Please check "Author Guidelines" for more information. <https://www.embopress.org/page/journal/17574684/authorguide#researcharticleguide>
- 8) Synopsis: Every published paper now includes a 'Synopsis' to further enhance discoverability. Synopses are displayed on the journal webpage and are freely accessible to all readers. They include separate synopsis image and synopsis text.
 - Synopsis image: Please provide a striking image or visual abstract as a high-resolution jpeg file 550 px-wide x (250-400)-px high to illustrate your article.
 - Synopsis text: Please provide a short standfirst (maximum of 300 characters, including space) as well as 2-5 one sentence bullet points that summarise the paper as a .doc file. Please write the bullet points to summarise the key NEW findings. They should be designed to be complementary to the abstract - i.e. not repeat the same text. We encourage inclusion of key acronyms and quantitative information (maximum of 30 words / bullet point). Please use the passive voice.
 - Please check your synopsis text and image before submission with your revised manuscript. Please be aware that in the proof stage minor corrections only are allowed (e.g., typos).
- 9) Source data: Source data for figure 3G are missing, please upload it.

10) For more information: This space should be used to list relevant web links for further consultation by our readers. Could you identify some relevant ones and provide such information as well? Some examples are patient associations, relevant databases, OMIM/proteins/genes links, author's websites, etc..

11) As part of the EMBO Publications transparent editorial process initiative (see our Editorial at <http://embomolmed.embopress.org/content/2/9/329>), EMBO Molecular Medicine will publish online a Review Process File (RPF) to accompany accepted manuscripts. This file will be published in conjunction with your paper and will include the anonymous referee reports, your point-by-point response and all pertinent correspondence relating to the manuscript. Let us know whether you agree with the publication of the RPF and as here, if you want to remove or not any figures from it prior to publication. Please note that the Authors checklist will be published at the end of the RPF.

12) Please provide a point-by-point letter INCLUDING my comments as well as the reviewer's reports and your detailed responses (as Word file).

I look forward to reading a new revised version of your manuscript as soon as possible.

Yours sincerely,

Zeljko Durdevic

*** Instructions to submit your revised manuscript ***

1) a .docx formatted version of the manuscript text (including Figure legends and tables)

2) Separate figure files*

3) supplemental information as Expanded View and/or Appendix. Please carefully check the authors guidelines for formatting Expanded view and Appendix figures and tables at <https://www.embopress.org/page/journal/17574684/authorguide#expandedview>

4) a letter INCLUDING the reviewer's reports and your detailed responses to their comments (as Word file).

5) The paper explained: EMBO Molecular Medicine articles are accompanied by a summary of the articles to emphasize the major findings in the paper and their medical implications for the non-specialist reader. Please provide a draft summary of your article highlighting

6) For more information: There is space at the end of each article to list relevant web links for further consultation by our readers. Could you identify some relevant ones and provide such information as well? Some examples are patient associations, relevant databases, OMIM/proteins/genes links, author's websites, etc...

7) Author contributions: the contribution of every author must be detailed in a separate section.

8) EMBO Molecular Medicine now requires a complete author checklist (<https://www.embopress.org/page/journal/17574684/authorguide>) to be submitted with all revised manuscripts. Please use the checklist as guideline for the sort of information we need WITHIN the manuscript. The checklist should only be filled with page numbers where the information can be found. This is particularly important for animal reporting, antibody dilutions (missing) and exact values and n that should be indicated instead of a range.

9) Every published paper now includes a 'Synopsis' to further enhance discoverability. Synopses are displayed on the journal webpage and are freely accessible to all readers. They include a short stand first (maximum of 300 characters, including space) as well as 2-5 one sentence bullet points that summarise the paper. Please write the bullet points to summarise the key NEW findings. They should be designed to be complementary to the abstract - i.e. not repeat the same text. We encourage inclusion of key acronyms and quantitative information (maximum of 30 words / bullet point). Please use the passive voice. Please attach these in a separate file or send them by email, we will incorporate them accordingly.

You are also welcome to suggest a striking image or visual abstract to illustrate your article. If you do please provide a jpeg file 550 px-wide x 300-800px high.

10) A Conflict of Interest statement should be provided in the main text

11) Please note that we now mandate that all corresponding authors list an ORCID digital identifier. This takes <90 seconds to complete. We encourage all authors to supply an ORCID identifier, which will be linked to their name for unambiguous name identification.

Currently, our records indicate that the ORCID for your account is 0000-0002-9708-1112.

Link Not Available

Photos 400-800 DPI

*Additional important information regarding figures and illustrations can be found at <https://bit.ly/EMBOPressFigurePreparationGuideline>. See also figure legend preparation guidelines: <https://www.embopress.org/page/journal/17574684/authorguide#figureformat>

***** Reviewer's comments *****

Referee #1 (Remarks for Author):

Mandolfo, et al submit a revised version of this manuscript, which reports findings in the mucopolysaccharidosis type IIIA (MPS IIIA) mouse model following challenge with poly Inositol / poly Cytidine (polyI:C), which stimulates the innate immune system and mimics the effects of infection. The purpose of the study is to explore the immune mechanisms which may underlie the neurodevelopmental decline often observed in people with MPS IIIA following acute intercurrent infections.

While the authors have taken great care to address both reviewers' concerns, there still remain significant issues with the manuscript which do not allow for publication in its current form.

REFEREE 1

Materials and Methods

1) Page 66 of 112 (Experimental Design) states 11 mice were used, but in the remainder of the manuscript, including figure 5, the maximum cohort size for any experiment was 10 mice. Please clarify.

2) Page 67 of 112 (High dose Intraperitoneal challenge / Low dose Intravenous challenge) does not align with the comments made in response to reviewers. Readers prefer to see symmetry between assays and correspondingly identical ages of study initiation and study termination; when it is not spelled out, the reading is more disjointed as they try to find parallels on their own. It is far more preferable for the manuscript to make that explicit to readers. Accordingly:

a) For Acute high dose, the methods were not entirely clear that there were actually 4 cohorts of 6 mice receiving high dose IP poly I:C. The referee initially thought that intracardiac puncture was performed on the same mice at 'saline,' +3h, +6h, and +24h to obtain the serum IL-1 β and TNF- α levels, but then realized that was not the case as brain histopathology from each time point indicates different mice were sacrificed to generate the serum cytokine and histopathology data. Please make it explicit that a separate cohort of six was euthanized at each time point, with four total cohorts used.

b) For Chronic high dose, the authors indicate in Response to Reviewers that mice were 2 months of age at first dose, and that the mice were sacrificed +12 weeks after the first dose.

However, the age of first dose is not specified in the methods (based upon the construction of the paragraph, which leads with acute high dose IP mice being dosed at 4 months of age, readers are left to infer that the age of dosing for chronic high dose is also 4 months). Please specify age of first dose = 2 months.

c) Further, the age of euthanasia for this group was not specified in methods. The text is constructed leading readers to infer possibly that the mice sacrificed 24h after the last (12 mg/kg) dose? Please specify time of euthanasia as +12 weeks.

d) For Chronic low dose, the methods does not state that the original intent was to assess the mice at +8 weeks (~ 4 months of age) and +12 weeks (~ 5 months of age), and the experiment was extended to +16 weeks (~ 6 months of age). Please explicitly state this, and explain (as you did in Response to Reviewers) why experiment was extended an additional 4 weeks.

e) In Materials and Methods, please explicitly state the weeks at which poly I:C were dosed for the two chronic assays (+1, +3, +9w for acute high; +1, +3, and +7w for acute low). Why did the timing of the last dose differ (+9w for high, +7w for low)? Your figure does a great job of explaining the study protocol; the text doesn't.

f) Why wasn't a time course (+3, +6, +24h) for Acute low dose poly I:C performed, paralleling the time course performed for Acute high dose poly I:C? Figures 3H-K only show data from +24h.

g) Why was brain caspase1 activity shown only for acute high dose (Figure 3G), but not for acute low dose? Likewise, why was hippocampal IL-1 β + cells shown only for acute low dose (Figure 3N) but not for acute high dose? Please explain. (i.e. please note in results that acute low dose poly I:C did not result in elevations of IL-1 β)

Results

Page 75 of 112 ("Interestingly, IL-1 β remained significantly up-regulated by 2.4 fold*s*...") 's' not necessary

Discussion

Page 84 of 112 ("This is in agreement with previous studies that showed that systemic infection*s* models, ...") 's' not necessary

REFeree 2

The explanation of poly(I:C)'s mechanism for stimulating (solely) the innate immune system is appreciated, as is the acknowledgment that an actual viral infection activates the immune system along multiple pathways. The limitations of poly(I:C) in simulating a viral infection are appreciated in the discussion.

In response to the steep dropoff of GFAP and NeuN signal so quickly after poly(I:C) dosing, suggest: "Challenge with an acute high poly(I:C) dose exacerbated systemic and brain cytokine expression, especially IL-1 β in the hippocampus. This was accompanied by an increase in caspase-1 activity within the brain of MPS IIIA mice with concomitant loss of hippocampal GFAP and NeuN expression."

Also, "... the underlying cause of this phenomenon primarily occurring in the hippocampus, and not in other brain regions, requires further investigation."

And, "The heightened inflammatory response induced by both acute and chronic poly(I:C) challenges may trigger metabolic perturbations in the MPS IIIA hippocampus ... This compromised astrocytic support may result in neuronal damage ..."

While it is true that the difference between IIIA saline and WT saline is ** and IIIA poly(I:C) and WT saline is ***, that comparison is not as meaningful as comparing IIIA saline to IIIA poly(I:C) - which is not significant.

The sentence "... and it [poly(I:C)] increased the significance of the response in MPS IIIA mice (Figure 3k)." is misleading readers who aren't looking at the figure directly into thinking there is a significant difference between IIIA saline and IIIA poly(I:C) TNFA. Please remove it.

Please introduce the abbreviation, 'FMO' (fluorescence minus one) as it was not explained either in text or figures.

Within your discussion, you assert there was neuronal death. While you show reduction of NeuN signaling, the authors still have not conclusively demonstrated nor confirmed cell death. The reviewer repeats from the first review, "Authors should reconsider their conclusion or should demonstrate/confirm cell death. More likely, the treatments affect gene expression, and/or other variables play a role in quantification of gene expression as performed in this study."

It is reiterated: "Please reconsider carefully."

Referee #2 (Remarks for Author):

The authors have addressed my comments. Please note a switch in figure labelling (3I/3H) between text and figures.

REFEREE 1:

MATERIALS AND METHODS

1) Page 66 of 112 (Experimental Design) states 11 mice were used, but in the remainder of the manuscript, including figure 5, the maximum cohort size for any experiment was 10 mice. Please clarify.

Some cohorts had 11 mice and others had 10 mice. This is due to some mix-ups in genotypes by our animal house at the allocation stage. We re-genotype at sacrifice to confirm status. We have therefore changed to n=10-11 to reflect this.

2) Page 67 of 112 (High dose Intraperitoneal challenge / Low dose Intravenous challenge) does not align with the comments made in response to reviewers. Readers prefer to see symmetry between assays and correspondingly identical ages of study initiation and study termination; when it is not spelled out, the reading is more disjointed as they try to find parallels on their own. It is far more preferable for the manuscript to make that explicit to readers. Accordingly:

a) For Acute high dose, the methods were not entirely clear that there were actually 4 cohorts of 6 mice receiving high dose IP poly I:C. The referee initially thought that intracardiac puncture was performed on the same mice at 'saline,' +3h, +6h, and +24h to obtain the serum IL-1b and TNF-a levels, but then realized that was not the case as brain histopathology from each time point indicates different mice were sacrificed to generate the serum cytokine and histopathology data. Please make it explicit that a separate cohort of six was euthanized at each time point, with four total cohorts used.

The high dose intraperitoneal challenge section has been edited accordingly:

For acute high dosing, 6 cohorts of 4-month-old female WT or MPS IIIA mice (n = 6 per group) were injected intraperitoneally with either 12 mg/kg of poly(I:C) in 200 µl saline³³ or saline as a control. Cohorts of 6 mice for poly(I:C) were measured for bodyweight and sacrificed at 3-, 6- and 24-hours post-injection by an intra-cardiac perfusion of PBS under terminal anaesthesia. The three saline controls were combined (n=18). The burrowing behaviour was assessed between 5- and 7-hours post-stimulation in the 6- and 24-hour cohorts.

For chronic high dosing, 4 cohorts of 2-month-old female WT or MPS IIIA mice were treated with either three injections of saline or three increasing doses of poly(I:C); 6 mg/kg (week 0; 2 months of age), 9 mg/kg (week 3) and 12 mg/kg (week 9) over a 9-week period (n = 10 per group). At 3-, 6- and 24-hours post-challenge, the body weight was measured. The burrowing behaviour was assessed between 5- and 7-hours post-stimulation. For both acute and chronic studies, blood was collected via cardiac puncture, allowed to clot at room temperature, centrifuged at 300 × g for 10 minutes, and serum stored at – 80 °C. At the end

of the chronic study, anaesthetised animals received an intra-cardiac perfusion of PBS and the tissues were harvested 12 weeks after the first injection.³²

b) For Chronic high dose, the authors indicate in Response to Reviewers that mice were 2 months of age at first dose, and that the mice were sacrificed +12 weeks after the first dose. However, the age of first dose is not specified in the methods (based upon the construction of the paragraph, which leads with acute high dose IP mice being dosed at 4 months of age, readers are left to infer that the age of dosing for chronic high dose is also 4 months). Please specify age of first dose = 2 months.

See above addition to clarify this.

c) Further, the age of euthanasia for this group was not specified in methods. The text is constructed leading readers to infer possibly that the mice sacrificed 24h after the last (12 mg/kg) dose? Please specify time of euthanasia as +12 weeks.

The following section was added to the high dose intraperitoneal challenge section: "At the end of the study, anaesthetised animals received an intra-cardiac perfusion of PBS and the tissues were harvested 12 weeks after first injection."³²

d) For Chronic low dose, the methods does not state that the original intent was to assess the mice at +8 weeks (~ 4 months of age) and +12 weeks (~ 5 months of age), and the experiment was extended to +16 weeks (~ 6 months of age). Please explicitly state this, and explain (as you did in Response to Reviewers) why experiment was extended an additional 4 weeks.

The following section was added to the low dose intravenous challenge: "The original intent for this study was to have a 12-week duration (in line with the high dose intraperitoneal study). However, since the behavioural analysis was not significant at 12 weeks, we extended the experiment length by 4 more weeks, to see an effect in our control groups."

e) In Materials and Methods, please explicitly state the weeks at which poly I:C were dosed for the two chronic assays (+1, +3, +9w for acute high; +1, +3, and +7w for acute low). Why did the timing of the last dose differ (+9w for high, +7w for low)? Your figure does a great job of explaining the study protocol; the text doesn't.

The weeks at which poly(I:C) were dosed for the two chronic assays were added. The last dosing time changed due to a mis-communication from the previous person performing the work – now clarified here.

f) Why wasn't a time course (+3, +6, +24h) for Acute low dose poly I:C performed, paralleling

the time course performed for Acute high dose poly I:C? Figures 3H-K only show data from +24h.

Once we knew the approximate time-course at which various cytokines responded from the high dose study, we no longer required a time course to capture the optimal response times for each cytokine. Typically cytokines respond in a set time-frame regardless of dose i.e. TNFalpha is quicker (3-6 hour peak) than IL1beta (24 hr peak) -but once we established that both were significantly elevated at 24 hours we were able to use this timepoint going forward – rather than have 6 cohorts of 6 mice.

g) Why was brain caspase1 activity shown only for acute high dose (Figure 3G), but not for acute low dose? Likewise, why was hippocampal IL-1 β + cells shown only for acute low dose (Figure 3N) but not for acute high dose? Please explain. (i.e. please note in results that acute low dose poly I:C did not result in elevations of IL-1 β) For acute low dose we decided not to perform the caspase-1 activity assay as we saw no significant elevation in hippocampal IL1-b+ cells, suggesting that we would not observe cell death. Serum IL1Beta does not necessarily tally with IL1Beta in specific regions, as it is typically active at picomolar amounts and also in pericellular locations, making it important to show which cells actually produce IL1B rather than just relying on serum levels. Thus, a non-significant change in serum IL1B could still mean destruction of brain tissue locally from an inflammasome mediated response. We did not quantify HS IL1beta in acute high dose, as the change was obvious.

RESULTS

Page 75 of 112 ("Interestingly, IL-1 β remained significantly up-regulated by 2.4 fold*s*...") 's' not necessary

The “s” was removed.

DISCUSSION

Page 84 of 112 ("This is in agreement with previous studies that showed that systemic infection*s* models, ...") 's' not necessary.

The “s” was removed.

REFEREE 2

In response to the steep dropoff of GFAP and NeuN signal so quickly after poly(I:C) dosing,

suggest: "Challenge with an acute high poly(I:C) dose exacerbated systemic and brain cytokine expression, especially IL-1 β in the hippocampus. This was accompanied by an increase in caspase-1 activity within the brain of MPS IIIA mice with concomitant loss of hippocampal GFAP and NeuN expression."

The relevant section in the abstract was edited according to the Referee's comment.

Also, "... the underlying cause of this phenomenon primarily occurring in the hippocampus, and not in other brain regions, requires further investigation."

The relevant section in the discussion was edited according to the Referee's comment.

And, "The heightened inflammatory response induced by both acute and chronic poly(I:C) challenges may trigger metabolic perturbations in the MPS IIIA hippocampus ... This compromised astrocytic support may result in neuronal damage ..."

The relevant section in the discussion was edited according to the Referee's comment.

While it is true that the difference between IIIA saline and WT saline is ** and IIIA poly(I:C) and WT saline is ***, that comparison is not as meaningful as comparing IIIA saline to IIIA poly(I:C) - which is not significant. The sentence "... and it [poly(I:C)] increased the significance of the response in MPS IIIA mice (Figure 3k)." is misleading readers who aren't looking at the figure directly into thinking there is a significant difference between IIIA saline and IIIA poly(I:C) TNFA. Please remove it.

We have removed this line

Please introduce the abbreviation, 'FMO' (fluorescence minus one) as it was not explained either in text or figures.

"Fluorescence minus one (FMO) controls are displayed" was added to Figure 4 legend.

Within your discussion, you assert there was neuronal death. While you show reduction of NeuN signaling, the authors still have not conclusively demonstrated nor confirmed cell death. The reviewer repeats from the first review, "Authors should reconsider their conclusion or should demonstrate/confirm cell death. More likely, the treatments affect gene expression, and/or other variables play a role in quantification of gene expression as performed in this study."

It is reiterated: "Please reconsider carefully."

Apologies we missed a couple of instances and we do agree with the reviewer. Changed accordingly from cell death to "cell damage" where referring to neurons – or apparent

astrocytic loss. “This was correlated with increased IL-1 β protein expression in the hippocampus, which may be localized to microglia-like cells, as well as exacerbation of caspase-1 activity and hippocampal **cell death**, all indicative of NLRP3 activation.” The word cell death has now been replaced with **cell damage**.

REFEREE 2:

Please note a switch in figure labelling (3I/3H) between text and figures.

Adjusted accordingly.

FROM THE JOURNAL

2) Authors: We note a discrepancy of author's name: Aiyin Liao in the manuscript file vs. Ai Yin Liao in our submission system. Please correct.

This was corrected accordingly.

3) In the main manuscript file, please do the following:

- Please address all comments suggested by our data editors listed below:

1. Please note that a separate 'Data Information' section is required in the legends of figures 1b-c, e-f; 2a-c; 3a-e, g-l, n; 4a-e; 6b-d; 7a-d; EV 2a-i; EV 4e-j.

Under each figure legend, we added the following statement: “The data relative to this figure is openly available at BioScience.”

2. Please note that the legends for figures 1b-e is not provided in the sequential manner (legend for figure 1d is provided before legends of figures 1b-c, legend for figure 1e is provided before legend of figure 1c). This needs to be rectified.

This was rectified.

3. Please note that the legend for figure 5c is not provided in the manuscript. This needs to be rectified.

This was rectified.

4. Please note that the legend for figure 8c is missing in the manuscript. This needs to be rectified.

This was rectified.

5. Please note that the legends for figures EV 2b-h is not provided in the sequential manner

(legends for figures EV 2d, g is provided before legends of figures 2b-c and 2b-f respectively, legend for figure 2e, h is provided before legends of figures 2c and 2c, f, i, respectively). This needs to be rectified.

This was rectified.

6. Please note that in figures 3a-e, g-l, n; 5b, f; 6b-d; EV 3a-h; there is a mismatch between the annotated p values in the figure legend and the annotated p values in the figure file that should be corrected.

This was rectified.

7. Please note that for the figure 8b, p-values and statistical tests are indicated in the legends. However, comparison for the same, ""*****/**/*/*"" has not been represented in the figure. Please rectify this in the figure or legend as applicable.

This was rectified.

8. Please note that information related to n is missing in the legends of figures 1b, e; 3n; EV 4b-d.

This was rectified.

9. Please note that n=2 in figures EV 4e-j.

Yes, it is correct. Added where missing in EV4.

- Correct order of the manuscript sections as follows: Abstract, Keywords, Introduction, Results, Discussion, Methods, Acknowledgements, Disclosure and competing interests statement, References, Figure legends, Tables and their legends, Expanded View Figure legends.

Rectified.

- Limit keywords to max 5.

Rectified.

- Please add callout for Fig 3I.

Added

- Statistical paragraph should reflect all information that you have filled in the Authors Checklist, especially regarding randomization, blinding, replication etc. Indicate in legends

exact p= values, not a range, along with the statistical test used. To keep the figures "clear" some authors found providing an Appendix table Sx with all exact p-values preferable. You are welcome to do this if you want to.

Some of the information required was initially placed in the Experimental design question. Now this has been added to the Statistical analysis section as follows: "Animals were recruited on staggered entry, as mice became old enough to be included, thus randomisation was not performed. However, mice were allocated equally across cohorts as recruitment progressed. It was impossible to blind treatment groups in cages, due to the nature of the mouse model and treatments given. Behavioural tests were recorded and analysed at a later time-point from video in a blinded fashion once all tests had been performed. Histochemical analyses were also carried out in a blinded fashion. Statistical analysis was performed using GraphPad Prism 9 software (La Jolla, CA, USA). Two-way analysis of variance (ANOVAs) were performed for multi-group analysis, followed by Tukey's multiple comparison test. Significance was set at P <0.05 %. The number of mice is stipulated by "n" in the respective figure legend. Data are presented as standard error of the mean (SEM) unless otherwise stated. Normality testing was performed and those not meeting these criteria were log transformed to normalise the dataset."

- Please rename "Conflict of interest" to "Disclosure Statement & Competing Interests". We updated our journal's competing interests policy in January 2022 and request authors to consider both actual and perceived competing interests. Please review the policy <https://www.embopress.org/competing-interests> and update your competing interests if necessary.

Rectified.

Please correct the reference citation in the text and reference list. In the text of the manuscript, a reference should be cited by author and year of publication. Include a space between a word and the opening parenthesis of the reference that follows. In the reference list, citations should be listed in alphabetical order. Where there are more than 10 authors on a paper, 10 will be listed, followed by "et al.". Please check "Author Guidelines" for more information. <https://www.embopress.org/page/journal/17574684/authorguide#referencesformat>

- Please provide data availability statement. If no data were deposited add the sentence "This study includes no data deposited in external repositories".

Rectified.

4) Remove ARRIVE Author Checklist.

Rectified.

5) Please upload Table EV1 and EV2 as separate files.

The Tables have now been added to the main manuscript as required as per manuscript order. The tables have also been uploaded separately.

6) Funding: Please merge it with "Acknowledgement. If the project number is available, please include it.

Rectified.

7) The Paper Explained: Please provide "The Paper Explained" and add it to the main manuscript text. Please check "Author Guidelines" for more information. <https://www.embopress.org/page/journal/17574684/authorguide#researcharticleguide>

This was added.

8) Synopsis: Every published paper now includes a 'Synopsis' to further enhance discoverability. Synopses are displayed on the journal webpage and are freely accessible to all readers. They include separate synopsis image and synopsis text.

- Synopsis image: Please provide a striking image or visual abstract as a high-resolution jpeg file 550 px-wide x (250-400)-px high to illustrate your article.
- Synopsis text: Please provide a short standfirst (maximum of 300 characters, including space) as well as 2-5 one sentence bullet points that summarise the paper as a .doc file. Please write the bullet points to summarise the key NEW findings. They should be designed to be complementary to the abstract - i.e. not repeat the same text. We encourage inclusion of key acronyms and quantitative information (maximum of 30 words / bullet point). Please use the passive voice.
- Please check your synopsis text and image before submission with your revised manuscript. Please be aware that in the proof stage minor corrections only are allowed (e.g., typos).

This was done and now provided.

9) Source data: Source data for figure 3G are missing, please upload it.

Unfortunately, we are unable to provide the source data for figure 3G as we cannot locate it.

10) For more information: This space should be used to list relevant web links for further consultation by our readers. Could you identify some relevant ones and provide such information as well? Some examples are patient associations, relevant databases, OMIM/proteins/genes links, author's websites, etc...

n/a

11) As part of the EMBO Publications transparent editorial process initiative (see our Editorial at <http://embomolmed.embopress.org/content/2/9/329>), EMBO Molecular Medicine will publish online a Review Process File (RPF) to accompany accepted manuscripts. This file will be published in conjunction with your paper and will include the anonymous referee reports, your point-by-point response and all pertinent correspondence relating to the manuscript. Let us know whether you agree with the publication of the RPF and as here, if you want to remove or not any figures from it prior to publication. Please note that the Authors checklist will be published at the end of the RPF.

We agree with the publication of the RPF.

4th Jun 2024

Dear Prof. Bigger,

We are pleased to inform you that your manuscript is accepted for publication and is now being sent to our publisher to be included in the next available issue of EMBO Molecular Medicine.
